# Correlative all-optical quantification of mass density and mechanics of subcellular compartments with fluorescence specificity

**Raimund Schlüßler[1]\*[†], Kyoohyun Kim[1,2]\*[†], Martin Nötzel[1], Anna Taubenberger[1], Shada Abuhattum[1,2], Timon Beck[1,2], Paul Müller[1,2], Shovamaye Maharana[1,3], Gheorghe Cojoc[1], Salvatore Girardo[1,2], Andreas Hermann[4], Simon Alberti[1,5], Jochen Guck[1,2,5]\***

[1]Biotechnology Center, Center for Molecular and Cellular Bioengineering, Technische Universität, Dresden, Germany; [2]Max Planck Institute for the Science of Light and Max-Planck-Zentrum für Physik und Medizin, Erlangen, Germany; [3]Department of Microbiology and Cell Biology, Indian Institute of Science, Bengaluru, India; [4]Translational Neurodegeneration Section "Albrecht Kossel", University Rostock, and German Center for Neurodegenerative Diseases (DZNE), Rostock/Greifswald, Germany; [5]Physics of Life, Technische Universität Dresden, Dresden, Germany

**\*For correspondence:**
raimund.schluessler@tu-dresden.de (RS);
kyoohyun.kim@mpl.mpg.de (KK);
jochen.guck@mpl.mpg.de (JG)

[†]These authors contributed equally to this work

**Competing interest:** The authors declare that no competing interests exist.

**Abstract** Quantitative measurements of physical parameters become increasingly important for understanding biological processes. Brillouin microscopy (BM) has recently emerged as one technique providing the 3D distribution of viscoelastic properties inside biological samples – so far relying on the implicit assumption that refractive index (RI) and density can be neglected. Here, we present a novel method (FOB microscopy) combining BM with optical diffraction tomography and epifluorescence imaging for explicitly measuring the Brillouin shift, RI, and absolute density with specificity to fluorescently labeled structures. We show that neglecting the RI and density might lead to erroneous conclusions. Investigating the nucleoplasm of wild-type HeLa cells, we find that it has lower density but higher longitudinal modulus than the cytoplasm. Thus, the longitudinal modulus is not merely sensitive to the water content of the sample – a postulate vividly discussed in the field. We demonstrate the further utility of FOB on various biological systems including adipocytes and intracellular membraneless compartments. FOB microscopy can provide unexpected scientific discoveries and shed quantitative light on processes such as phase separation and transition inside living cells.

## Editor's evaluation

This is an important and timely contribution that introduces a new approach that combines Brillouin microscopy with fluorescence (FOB) to measure the mechanical properties in terms of longitudinal moduli for viscoelastic materials in cells. This approach has many promising applications, which the authors articulate, and could be important as a new and complementary modality for investigating the mechanical properties of soft materials, specifically membrane-bound and membrane-less organelles.

## Introduction

The mechanical properties of tissues, single cells, and intracellular compartments are linked to their function, in particular during migration and differentiation, and as a response to external stress (*Engler et al., 2006*; *Provenzano et al., 2006*; *Lo et al., 2000*). Hence, characterizing mechanical properties in vivo has become important for understanding cell physiology and pathology, for example, during development or cancer progression (*Mammoto et al., 2013*; *Jansen et al., 2015*; *Mohammed et al., 2019*). To measure the mechanical properties of biological samples, many techniques are available. These include atomic force microscopy (*Christ et al., 2010*; *Koser et al., 2015*; *Gautier et al., 2015*; *Franze et al., 2013*), micropipette aspiration (*Maitre et al., 2012*), and optical traps (*Wu et al., 2018a*; *Litvinov et al., 2002*; *Bambardekar et al., 2015*; *Guck et al., 2001*). These techniques can access the rheological properties of a sample and their changes under various pathophysiological conditions. Yet, most of them require physical contact between probe and sample surface and none of them allows to obtain spatially resolved distributions of the mechanical properties inside the specimens.

Brillouin microscopy has emerged as a novel microscopy technique to provide label-free, noncontact, and spatially resolved measurements of the mechanical properties inside biological samples (*Scarcelli and Yun, 2008*; *Scarcelli et al., 2015*; *Antonacci et al., 2018*; *Prevedel et al., 2019*). The technique is based on Brillouin light scattering that arises from the inelastic interaction between the incident photons and collective fluctuations of the molecules (acoustic phonons) (*Brillouin, 1922*; *Boyd, 2008*). The Brillouin shift measured is related to the longitudinal modulus, refractive index (RI), and absolute density, and the Brillouin peak linewidth is associated with the viscosity of the sample (see Materials and methods). The longitudinal modulus characterizes the compressibility of a sample and is in the GPa range for common biological samples (*Prevedel et al., 2019*). While the longitudinal modulus is theoretically related to the more commonly used Young's modulus by the Poisson's ratio, a conversion between the two moduli is generally not possible since the Poisson's ratio is frequency dependent and normally unknown. However, multiple studies found empirical correlations between the longitudinal modulus and the Young's modulus (*Scarcelli et al., 2011*; *Scarcelli et al., 2015*; *Schlüßler et al., 2018*). Furthermore, the longitudinal modulus takes into account all instrument properties like wavelength or scattering angle, and does not need normalization for comparability between different setups as the Brillouin shift does (*Antonacci et al., 2020*). So far, conventional Brillouin microscopy does not consider the contribution of heterogeneous RI and absolute density distributions to the longitudinal modulus. Most studies either assume a homogeneous RI distribution (*Scarcelli and Yun, 2008*; *Scarcelli et al., 2011*; *Antonacci and Braakman, 2016*), argument that the RI and absolute density trivially cancel out (*Scarcelli et al., 2012*; *Scarcelli et al., 2015*; *Antonacci et al., 2018*), or use RI values obtained separately by other imaging setups (*Schlüßler et al., 2018*). Other approaches to calculate the longitudinal modulus measure the mass density of the sample, but still rely on a priori knowledge of the RI (*Liu et al., 2019*; *Remer et al., 2020*). These simplifications may result in an inaccurate calculation of the longitudinal modulus since the RI distribution might not be homogenous throughout the sample, RI and density might not be coupled, hence, not cancel out, or the sample preparations necessary for separate RI measurements could influence the RI measured. Only recently serial Brillouin measurements of samples illuminated under different illumination angles allowed measuring the RI value inside the focal volume as well (*Fiore et al., 2019*). However, this technique requires illuminating the sample from two different directions, which doubles the acquisition time and decreases the spatial resolution of the measurement when compared to a setup only acquiring the Brillouin shift.

Optical diffraction tomography (ODT) has been utilized for measuring the three-dimensional (3D) RI distribution of various specimens (*Sung et al., 2009*; *Cotte et al., 2013*; *Kim et al., 2016a*). Employing quantitative phase imaging, ODT can reconstruct the 3D RI distribution of living biological samples from the complex optical fields measured under different illumination angles. Given the RI, the mass density and protein concentration of most biological samples can be calculated using a two-substance mixture model (see Materials and methods) (*Barer, 1952*; *Popescu et al., 2008*; *Zangle and Teitell, 2014*). Protein concentrations acquired with ODT were shown to agree well with results from volume-based measurements and did not suffer from differences in the quantum yield of fluorescent dyes between dilute and condensed phase as it might happen for fluorescence intensity ratio measurements (*McCall et al., 2020*). However, using the two-substance mixture model requires knowledge of the refraction increment, which depends on the material composition and takes on

values of 0.173-0.215 ml/g  with an average of 0.190 ml/g for different human proteins (**Zhao et al., 2011**; **Theisen, 2000**) and can go down to 0.135-0.138 ml/g  for phospholipids (**Erbe and Sigel, 2007**; **Mashaghi et al., 2008**). Furthermore, the two-substance mixture model does not apply to cell compartments mainly filled with a single substance, for example, lipid droplets in adipocytes. Hence, specificity by, for example, fluorescence imaging is necessary to determine whether the two-substance mixture model is appropriate and which refraction increment should be used to calculate the absolute density of a certain cell region.

Here, we present a combined optical system for epifluorescence, ODT, and Brillouin microscopy (FOB microscopy), which can provide the correct longitudinal modulus from colocalized measurements of the Brillouin shift and RI distributions and the subsequently calculated absolute densities of a sample. The principal function of the FOB microscope is demonstrated by measurements of cell phantoms made of biconstituent polymers with known mechanical properties. We further applied the setup to HeLa cells and adipocytes. First, we investigated two condensates that form by physical process of phase separation – nucleoli in the nucleus and stress granules (SGs) in the cytoplasm (**Alberti and Dormann, 2019**). Nucleoli in HeLa cells showed a higher RI ($n = 1.3618 \pm 0.0004$) and longitudinal modulus ($M' = 2.487 \pm 0.005\,\text{GPa}$) than the cytoplasm ($n = 1.3545 \pm 0.0004$, $M' = 2.410 \pm 0.005\,\text{GPa}$), whereas the nucleoplasm had a lower RI ($n = 1.3522 \pm 0.0004$) than the cytoplasm while still showing a higher longitudinal modulus ($M' = 2.448 \pm 0.005\,\text{GPa}$). The RI of the cytoplasm and nucleoplasm decreased after stressing HeLa cells with arsenite, but we found no statistically significant difference of either the RI or longitudinal modulus of SGs to the surrounding cytoplasm. By contrast, poly-Q aggregates formed by overexpressing the aggregation-prone exon 1 of Q103 huntingtin exhibited a 2.5% higher RI and 20.0% higher longitudinal modulus compared to the surrounding compartment. Moreover, unlike water-based cellular condensates and aggregates, lipid droplets inside adipocytes showed higher RI and Brillouin shift, but lower longitudinal modulus than the cytoplasm when taking into account their absolute density. These data illustrates that in order to correctly calculate the longitudinal modulus the RI as well as the absolute density have to be taken into account. In summary, the presented setup could provide measurement data necessary for a deeper understanding of pathophysiological processes related to cell mechanics and condensates that form by the process of phase separation.

## Results
### Optical setup

FOB microscopy combines ODT with Brillouin microscopy and epifluorescence imaging (**Figure 1a**). The three imaging modalities are sequentially applied to quantitatively map the RI, the Brillouin shift, and the epifluorescence intensity distribution inside a given sample. These parameters allow to, for example, infer the mass density and dry mass of the sample, and provide specificity to fluorescently labeled structures. Given the fluorescence specificity, it is furthermore possible to localize subcellular organelles of interest and to determine whether for a certain region the two-substance mixture model can be applied to calculate the local absolute density, or if the literature value of the absolute density has to be used (e.g., in lipid droplets). Finally, with the combination of RI, absolute density, and Brillouin shift distributions, the longitudinal modulus can be calculated.

For ODT, the sample is illuminated with a plane wave under different incident angles. To illuminate the sample under different angles, a dual-axis galvanometer mirror tilts the illumination beam. The transmitted light interferes with a reference beam and creates a spatially modulated hologram on a camera from which the phase delay and finally the RI of the sample are calculated with a resolution of 0.25 µm within the lateral plane and 0.5 µm in the axial direction (**Figure 1c**). Epifluorescence microscopy captures the fluorescence emission intensity image (**Figure 1b**) with the same camera used for the ODT acquisition.

For Brillouin microscopy, a moveable mirror guides the incident light to an additional lens, which leads to a focus in the sample with a size of 0.4 µm in the lateral plane and approximately 1 µm in axial direction. The focus is translated by the galvanometer mirror to scan the whole sample. The Brillouin scattered light is collected in the backscattering configuration and guided to a two-stage virtually imaged phased array (VIPA) spectrometer (**Scarcelli and Yun, 2008**). The Brillouin shift (**Figure 1d**) is

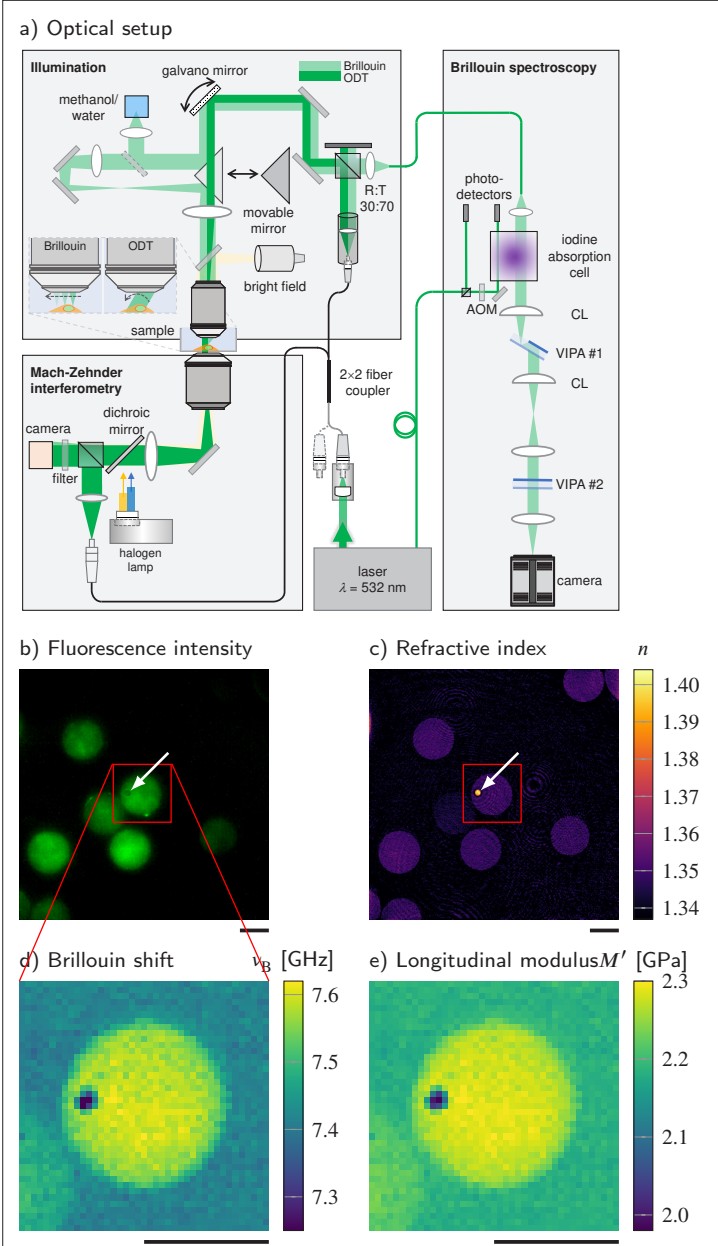

**Figure 1.** Combined fluorescence, optical diffraction tomography (ODT), and Brillouin microscopy. (**a**) Optical setup. The beam paths for epifluorescence/brightfield imaging, ODT, and Brillouin microscopy are shown in light yellow, dark green, and light green, respectively. The laser light illuminating the sample is collimated in ODT mode and focused in Brillouin mode. A moveable mirror enables to switch between the two modes. The Brillouin scattered light is guided to the spectrometer by a single-mode fiber, which acts as confocal pinhole. The light transmitted through the sample interferes with a reference beam. AOM, acousto-optic modulator; CL, cylindrical lens; LED, light-emitting diode; VIPA, virtually imaged phased array. (**b–e**) Quantitative and spatially resolved maps of a cell phantom consisting of a polydimethylsiloxane (PDMS) bead (indicated by the white arrows) inside a polyacrylamide (PAA) bead stained with Alexa 488 (green fluorescence in **b**) acquired with the FOB microscope. (**b**) Epifluorescence intensities, (**c**) refractive indices, (**d**) Brillouin shifts, and (**e**) calculated longitudinal moduli. The size of the Brillouin map is 41 × 41 pixel, resulting in an acquisition duration of 30 min. Scale bars 10 μm.

The online version of this article includes the following figure supplement(s) for figure 1:

**Figure supplement 1.** Absolute density of a cell phantom consisting of a polydimethylsiloxane (PDMS) bead inside a polyacrylamide (PAA bead).

extracted from the recorded Brillouin spectrum, and the longitudinal modulus (*Figure 1e*) is calculated from the Brillouin shift, RI, and absolute density distributions acquired (see Materials and methods).

## Validation of the setup with cell phantoms

To validate the basic performance of the combined FOB microscopy setup, we acquired the RI and Brillouin shift of an artificial cell phantom with known material properties. The phantom consists of a polydimethylsiloxane (PDMS) bead embedded in a polyacrylamide (PAA) bead (*Figure 1b–e*), which was fluorescently labeled with Alexa 488 (see Materials and methods). The material properties of the two components of the phantom are expected to be homogeneous, so that the standard deviation (SD) of the values measured can be used as an estimate of the setups' measurement uncertainty. The RI of the embedded PDMS bead was measured as 1.3920 ± 0.0080 (mean value ± SD) (*Figure 1c*). This was slightly lower than the values reported for bulk PDMS with the RI of 1.416 (*Meichner et al., 2015*), which could be due to the swelling of the PDMS beads during the fabrication process (*Wang et al., 2015*). The RI of the PAA bead (1.3485 ± 0.0024) was substantially lower than that of the PDMS bead and was close to the previously reported value (*Girardo et al., 2018*). In contrast, the Brillouin shift of the PDMS bead (7.279 ± 0.043 GHz) was lower than for the PAA bead (7.574 ± 0.020 GHz) (*Figure 1d*). In order to calculate the longitudinal modulus, the absolute density of the PAA bead (1.019 ± 0.001 g/ml) was calculated from the measured RI by applying a two-substance mixture model (see Materials and methods). However, this model cannot be applied for the PDMS bead since the bead does not contain a fluid phase. Hence, the area of the PDMS bead was segmented based on the RI and fluorescence intensity (*Figure 1b*), and the literature value for the absolute density of PDMS (1.03 g/ml) was used (see *Figure 1—figure supplement 1*; *Rahman et al., 2012*). The resulting

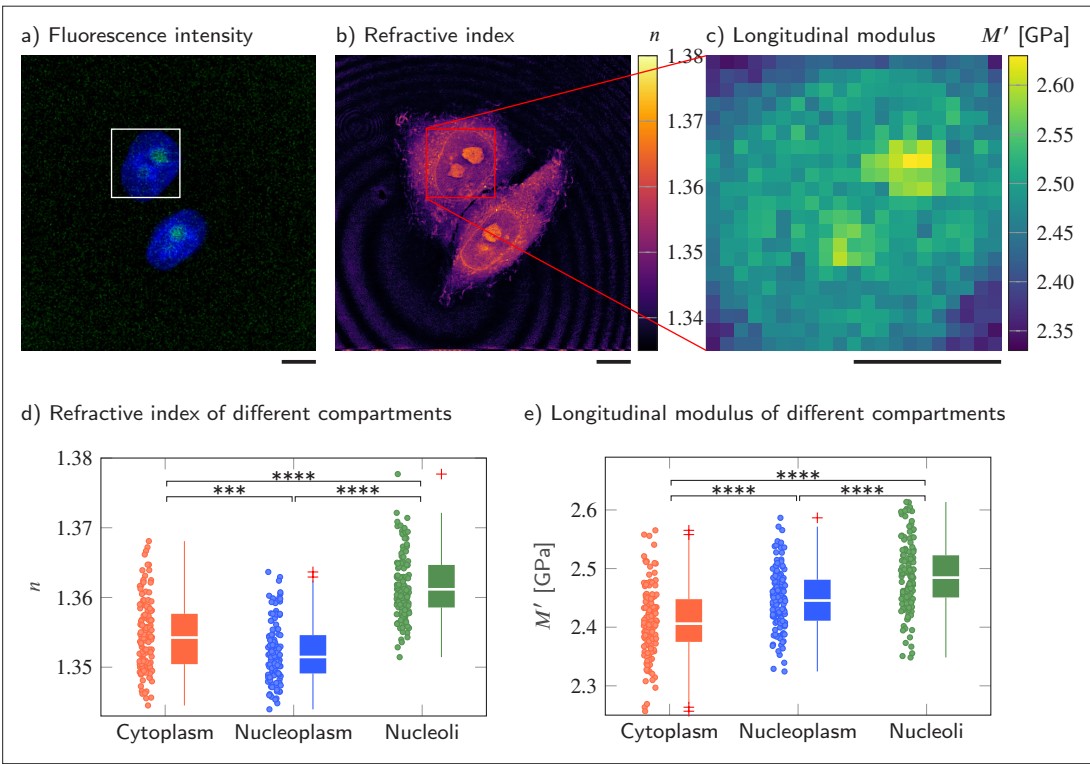

**Figure 2.** Cell nucleoplasm has lower refractive index (RI) but higher longitudinal modulus than cytoplasm. (**a–c**) Representative maps of the (**a**) epifluorescence intensity distribution, (**b**) longitudinal moduli, and (**c**) RIs of a HeLa cell. Nuclei are stained with Hoechst (blue), and the nucleolar protein in the nucleoli is labeled with GFP (green). Quantitative analysis of (**d**) the RI and (**e**) the calculated longitudinal modulus taking into account the Brillouin shifts, RIs, and absolute densities of 139 HeLa cells. The size of the Brillouin map is 21 × 21 pixel, resulting in an acquisition duration of 8 min . Scale bars 10 µm. \*\*\*p<0.001, \*\*\*\*p<0.0001.

The online version of this article includes the following source data and figure supplement(s) for figure 2:

**Source data 1.** Refractive index, longitudinal modulus, Brillouin shift, and absolute density values of different compartments in HeLa cells.

**Figure supplement 1.** Brillouin shift and absolute density of cytoplasm, nucleoplasm, and nucleoli in HeLa cells.

longitudinal modulus is shown in *Figure 1e*. We found values of 2.022 ± 0.030 GPa for the PDMS bead and 2.274 ± 0.012 GPa for the PAA bead. The results are consistent with previous measurements of the speed of sound in PDMS (*Cafarelli et al., 2017*) and the longitudinal modulus of PAA (*Schlüßler et al., 2018*) when taking into account the absolute density of the dry fraction (i.e., (*Equation 2*)). Our finding clearly demonstrates the strength of the presented FOB setup to provide local RI and absolute density distributions for correctly calculating longitudinal modulus from the Brillouin shift measured.

## Cell nucleoplasm has lower absolute density but higher longitudinal modulus than cytoplasm

The FOB microscope can also provide the much needed quantitative insight into a biological phenomenon that has recently captured the imagination of physicists and biologists alike – the formation of membraneless compartments by liquid-liquid phase separation (LLPS) (*Brangwynne et al., 2011*). One such membraneless compartment is the nucleolus, a region within the nucleus where ribosomal subunits are synthesized. Here, we recorded the epifluorescence, Brillouin shift, and RI distributions of 139 HeLa cells in which a nucleolar marker protein NIFK was tagged with GFP and the nuclei were stained with Hoechst (see Materials and methods). In order to evaluate the mechanical properties of the cytoplasm, nucleoplasm, and nucleoli separately, we segmented the different compartments of the cells based on the RI and the two-channel epifluorescence intensity maps (*Figure 2a*, see Materials and methods).

As shown in *Figure 2b and d*, the nucleoplasm of HeLa cells exhibited a statistically significantly lower RI than the cytoplasm (Kruskal−Wallis $p_{n_{\mathrm{cyto}},n_{\mathrm{np}}} = 9 \times 10^{-4}$), with values of 1.3522 ± 0.0004 (mean value ± SEM) (nucleoplasm) and 1.3545 ± 0.0004 (cytoplasm), which is consistent with previous studies (*Schürmann et al., 2016*; *Kim and Guck, 2020*). Since the RI of the HeLa cells measured is linearly proportional to their mass density (*Kim and Guck, 2020*), we applied the two-substance mixture model and used a global refraction increment of 0.190 ml/g, which is valid for protein and nucleic acid (*Zhao et al., 2011*; *Zangle and Teitell, 2014*), to calculate the absolute densities of each cell and its compartments. The resulting absolute densities are shown in *Figure 2—figure supplement 1b and d*. We found that the nucleoplasm had a lower absolute density (1.0207 ± 0.0005 g/ml) than the cytoplasm (1.0234 ± 0.0006 g/ml). Here, the perinuclear cytoplasm also contains many lipid-rich membrane-bound organelles, and the RI increment of phospholipids (0.135-0.138 ml/g, *Erbe and Sigel, 2007*; *Mashaghi et al., 2008*) is lower than that of protein and nucleic acid. Hence, the calculated absolute density of the cytoplasm could be underestimated and the absolute density difference between cytoplasm and nucleoplasm might be even more pronounced.

Interestingly, the Brillouin shift of the nucleoplasm (7.872 ± 0.007 GHz) was statistically significantly higher than the value of the cytoplasm (7.811 ± 0.008 GHz) ($p_{\nu_{\mathrm{B,cyto}},\nu_{\mathrm{B,np}}} = 2 \times 10^{-6}$, *Figure 2—figure supplement 1b and d*). The longitudinal moduli of the nucleoplasm (2.448 ± 0.005 GPa) and cytoplasm (2.410 ± 0.005 GPa) followed the same trend as the Brillouin shifts ($p_{M'_{\mathrm{cyto}},M'_{\mathrm{np}}} = 7 \times 10^{-7}$, *Figure 2c and e*). Moreover, the nucleoli, where ribosomal subunits are synthesized, exhibited statistically significantly higher RI ($n = 1.3618 \pm 0.0004$), Brillouin shift ($\nu_B = 7.938 \pm 0.008\,\mathrm{GHz}$), and longitudinal modulus ($M' = 2.487 \pm 0.005\,\mathrm{GPa}$) than either nucleoplasm or cytoplasm. We further found that the Brillouin peak linewidth of the cytoplasm and nucleoplasm is not statistically significantly different, but shows a statistically significant increase in the nucleoli (*Figure 2—figure supplement 1*). This indicates a higher viscosity and a less fluid-like behavior in the nucleoli compared to the cytoplasm and nucleoplasm. A full list of the resulting RI, Brillouin shifts, absolute densities, longitudinal moduli and linewidths, and the corresponding p-values when comparing between different cell compartments can be found in *Supplementary files 1 and 2*.

These findings imply that membraneless compartments formed by phase separation, in this case the nucleolus, can maintain a higher absolute density and distinct compressibility (here, higher longitudinal modulus) in spite of the thermodynamic instability inherent in this state.

## polyQ aggregates have higher absolute density and longitudinal modulus than cytoplasm

To compare the properties of physiological condensates with a densely packed protein aggregate, we overexpressed an expanded version of the aggregation-prone exon 1 of huntingtin with 103 consecutive glutamines (*Lieberman et al., 2019*; *Norrbacka et al., 2019*; *Bäuerlein et al., 2017*). Q103

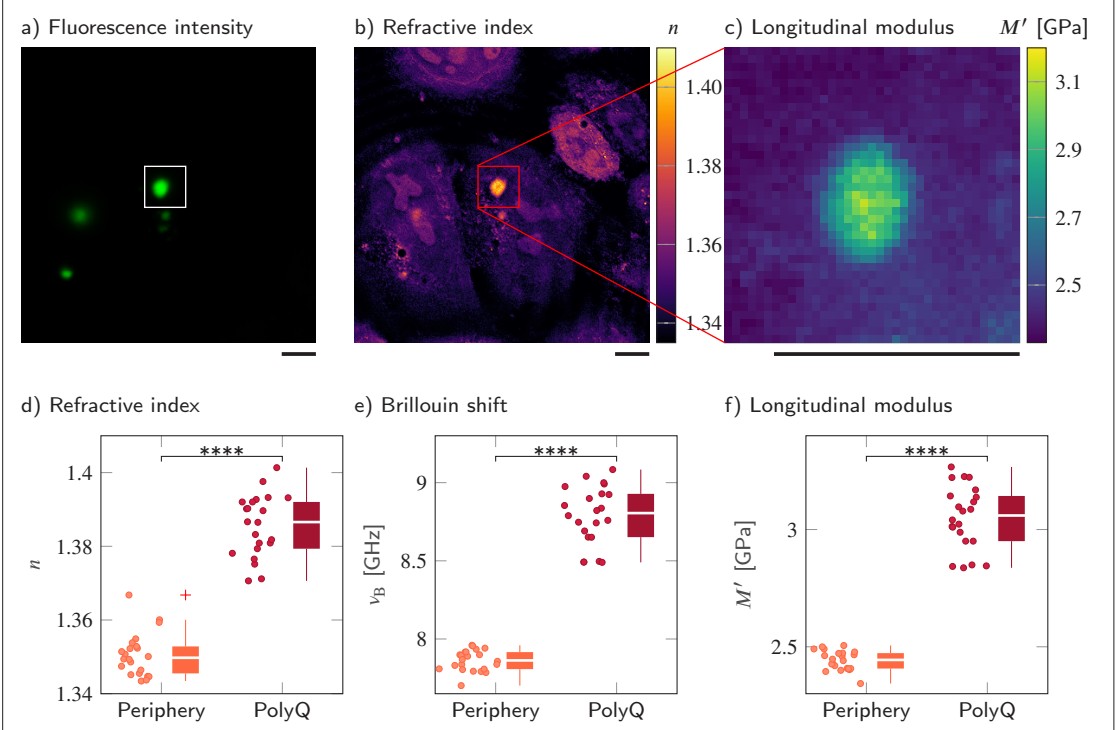

**Figure 3.** Polyglutamine (polyQ) aggregates have a higher refractive index, Brillouin shift, and longitudinal modulus than the peripheral cytoplasm. (**a–c**) Representative maps of (**a**) the epifluorescence intensity distribution, (**b**) the refractive indices, and (**c**) the longitudinal moduli of a HeLa cell transfected with a plasmid encoding HttQ103. The polyQ aggregates are labeled with GFP (green). Quantitative analysis of (**d**) the refractive index, (**e**) the Brillouin shift, and (**f**) the calculated longitudinal modulus taking into account the Brillouin shifts, refractive indices, and absolute densities of 22 polyQ granules. The size of the Brillouin map is 37 × 37 pixel, resulting in an acquisition duration of 23 min. Scale bars µm. ****$p<0.0001$.

The online version of this article includes the following source data for figure 3:

**Source data 1.** Refractive index, Brillouin shift, and longitudinal modulus values of polyglutamine (polyQ) aggregates and their periphery.

phase separates into liquid droplets in cells and these droplets rapidly convert into a solid-like state (*Yang and Yang, 2020*), meaning that they do not recover from photobleaching when subjected to fluorescence recovery after photobleaching (FRAP) experiments (*Kroschwald et al., 2015*). Here, we observe polyglutamine (polyQ) aggregates labeled with GFP in transiently transfected wild-type HeLa cells. We used the FOB microscope to measure the mechanical properties of polyQ granules in 22 different cells.

The polyQ aggregates showed a strong fluorescence signal in the GFP channel (*Figure 3a*). We hence used the fluorescence intensity to segment the aggregates from the peripheral cytoplasm in order to quantitatively compare cytoplasm and aggregates (*Figure 3b and c*). The RI (1.3856 ± 0.0018) and the longitudinal modulus (3.051 ± 0.029 GPa) of the aggregates were statistically significantly higher ($p<0.0001$) than the RI (1.3506 ± 0.0013) and longitudinal modulus (2.442 ± 0.009 GPa) of the peripheral cytoplasm (*Figure 3d and f* and *Supplementary file 3*). Using the RI measured, we find a protein concentration of 255.8 ± 9.4 mg/ml in the polyQ aggregates, a fourfold higher concentration than the value of 65 mg/ml previously measured with ODT in a G3BP1 in vitro system (*Guillén-Boixet et al., 2020*). Our results show that FOB microscopy can quantify the physical properties of cytoplasmic membraneless condensates – in principle.

## FUS-GFP SGs in living P525L HeLa cells show RI and longitudinal modulus similar to the surrounding cytoplasm

Recently, another type of condensates formed by LLPS – RNA and protein (RNP) granules, such as SGs – has received much attention due to its linkage to neurodegenerative diseases such as amyotrophic lateral sclerosis (ALS) and frontotemporal dementia (*Patel et al., 2015*; *Alberti and Hyman, 2016*). It is also increasingly recognized that the mechanical properties of these compartments influence their

functions and involvement in disease (*Jawerth et al., 2018*; *Nötzel et al., 2018*). Fused in sarcoma (FUS) protein, an RNA-binding protein involved in DNA repair and transcription, is one example of a protein that localizes to SGs (*Patel et al., 2015*). Purified FUS protein is able to phase separate into liquid condensates in vitro, and this property is important for FUS to localize to SGs. Disease-linked mutations in FUS have been shown to promote a conversion of reconstituted liquid FUS droplets from a liquid to a solid state, suggesting that an aberrant liquid to solid transition of FUS protein promotes disease.

Conventionally, the mechanical changes of SGs have been indirectly characterized by FRAP or observing fusion events of liquid droplets (*Brangwynne et al., 2009*). Recently, Brillouin microscopy was used to measure the Brillouin shift of SGs in chemically fixed P525L HeLa cells expressing mutant RFP-tagged FUS under doxycycline exposure (*Antonacci et al., 2018*). P525L HeLa cells are used as a disease model for ALS and form SGs under arsenite stress conditions. It was shown that the Brillouin shift of SGs induced by arsenite treatment with mutant RFP-FUS is statistically significantly higher than the Brillouin shift of SGs without mutant RFP-FUS. Furthermore, the Brillouin shift of mutant RFP-FUS SGs was reported to be statistically significantly higher than the value of the surrounding cytoplasm (*Antonacci et al., 2018*).

Here, we applied the FOB setup to P525L HeLa cells that express GFP-tagged FUS and quantified the RI distributions, epifluorescence intensities, and Brillouin shifts of the nucleoplasm, cytoplasm, and SGs. As not all HeLa cells were GFP-positive (see *Figure 4a*), we only selected the ones with a clear signal in the GFP channel. The cells were measured under control conditions after oxidative stress conditions when exposed to 5 mM sodium arsenite $NaAsO_2$ 30 min prior to the measurements and after chemical fixation after oxidative stress. Since the SGs are not static, and assemble and disassemble dynamically in living cells, acquiring the Brillouin shift map of a complete cell would be too slow, which was the reason for the chemical fixation of the cells in a previous study (*Antonacci et al., 2018*). During the approximate duration of 20-30 min of a whole-cell measurement, SGs moved substantially or even disassembled and, hence, did not colocalize with their epifluorescence signal acquired before. Furthermore, the P525L FUS-GFP HeLa cells reacted sensitively to the exposure to green laser light and suffered from cell death during a whole-cell measurement. We therefore did not acquire Brillouin shift maps of complete P525L HeLa cells, but only of small regions of 5 μm × 5 μm around the SGs or the corresponding regions in the cytoplasm of the control cells. This reduced the measurement duration to less than 2 min, allowing us to colocalize the SGs Brillouin shift and epifluorescence signal and preventing cell death during the acquisition (see *Figure 4—figure supplement 1*). Hence, all measurements presented here stem from living cells. The positions for the Brillouin shift measurements of the different compartments were chosen manually based on the epifluorescence and brightfield intensities (see *Figure 4a–f*). In total, we measured over 100 different cells, with the number of values per compartment and condition varying from 32 to 42, as shown in *Figure 4g and h*.

We found that the RI of the cytoplasm measured in the region of SG formation was statistically significantly lower than the RI of the nucleoplasm for all conditions (control [c], arsenite [a], and arsenite fixed [f]) tested ($p_c = 0.034$, $p_a = 0.035$, $p_f = 5 \times 10^{-6}$, *Figure 4g* and *Supplementary file 4*). However, when segmenting the RI of the whole cell and not only taking into account the RI of the manually selected regions for which we also performed measurements of the Brillouin shift, we found a slightly, although not statistically significantly, lower RI in the nucleoplasm than in the cytoplasm (*Figure 4—figure supplement 2*). Hence, we think the higher RI of the nucleoplasm is a result of the manual selection of the measurement positions in the region of SG formation near the cell boundary. As for wild-type HeLa cells, the longitudinal modulus of the nucleoplasm was statistically significantly higher than the modulus of the cytoplasm for all conditions ($p_c = 1 \times 10^{-4}$, $p_a = 8 \times 10^{-6}$, $p_f = 1 \times 10^{-11}$, *Figure 4h* and *Supplementary file 4*). While the GFP-tagged FUS of the control cells was mainly located in the nucleoplasm (*Figure 4a*), after arsenite treatment, the FUS was relocated from the nucleoplasm and aggregated in SGs within the cytoplasm (*Figure 4b*). This was accompanied by a statistically significant decrease of the RI of both the peri-SG cytoplasm ($n_{a,peri-cyto} = 1.3456 \pm 0.0007$, $p = 0.019$) and the nucleoplasm ($n_{a,nucleo} = 1.3476 \pm 0.0006$, $p = 0.017$) as well as of the longitudinal modulus of the peri-SG cytoplasm ($M'_{a,peri-cyto} = 2.329 \pm 0.009 \, GPa$, $p = 0.036$). Furthermore, we found no statistically significant difference of neither the RI ($n_{a,SG} = 1.3443 \pm 0.0006$) nor the longitudinal modulus ($M'_{a,SG} = 2.345 \pm 0.009 \, GPa$) of SGs to the respective values in the peri-SG cytoplasm. However, after chemical fixation the longitudinal

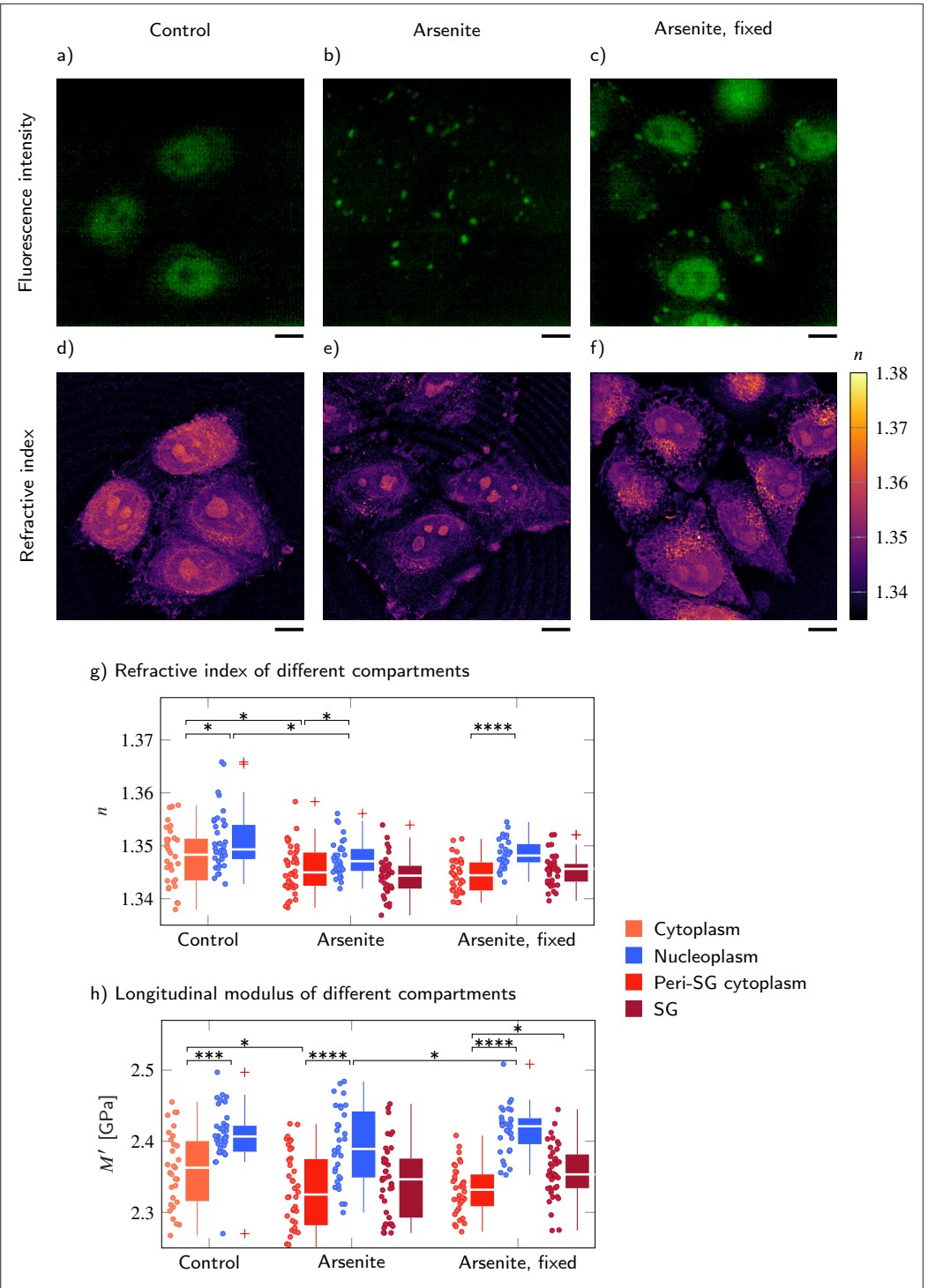

**Figure 4.** FUS-GFP-labeled stress granules induced by oxidative stress in living P525L FUS HeLa cells show a similar refractive index (RI), and longitudinal modulus as the peripheral cytoplasm. Representative example of (**a–c**) the fluorescence intensity and (**d–f**) the RI distribution under control conditions without arsenite, with arsenite, and with arsenite after fixation, respectively. Quantitative analysis of (**g**) the RI and (**h**) the calculated longitudinal modulus taking into account the Brillouin shift and RI. Scale bars 10 μm. *$p < 0.05$, **$p < 0.01$, ***$p < 0.001$, ****$p < 0.0001$.

The online version of this article includes the following source data and figure supplement(s) for figure 4:

**Source data 1.** Refractive index and longitudinal modulus values of cytoplasm, nucleoplasm, and stress granules

(SGs) in P525L HeLa cells after different treatments.

**Figure supplement 1.** Comparison of the influence of different acquisition schemes and laser wavelengths on the viability of P525L HeLa cells that express GFP-tagged FUS.

**Figure supplement 2.** Evaluation of the refractive index (RI) of P525L FUS HeLa cells taking into account the complete cell.

---

modulus of the SGs ($M'_{\text{f,SG}} = 2.357 \pm 0.006\,\text{GPa}$) was statistically significantly higher than the longitudinal modulus of the cytoplasm ($M'_{\text{f,peri-cyto}} = 2.331 \pm 0.006\,\text{GPa}$, $p = 0.030$).

Altogether, in P525L HeLa cells expressing FUS-GFP, the RI and longitudinal modulus of the nucleoplasm of the control, arsenite-treated, and fixed cells was statistically significantly higher than the respective values in the cytoplasm. Interestingly, SGs showed no statistically significant differences to the peri-SG cytoplasm in living, arsenite-treated cells, but had a statistically significantly higher longitudinal modulus in arsenite-treated and chemically fixed cells. This is consistent to previous studies showing a higher longitudinal modulus of SGs compared to the cytoplasm in chemically fixed P525L HeLa cells (*Antonacci et al., 2018*) and that fixation can substantially alter the mechanical (*Braet et al., 1998*) as well as the optical properties (*Su et al., 2014*) of biological samples.

## Mechanical characterization of lipid droplets in adipocytes requires precise RI and density

Most biological cells can be thought of as a mixture of ions and macromolecules such as proteins, nucleic acids, and sugars dissolved in water, for which the two-substance mixture model (*Barer, 1952*; *Popescu et al., 2008*; *Zangle and Teitell, 2014*) is appropriate to describe the relationship between the RI and the absolute density. However, this is not the case for special compartments in certain cell types. The lipid droplets within adipocytes are not composed of a water-based solution and cannot be characterized by the two-substance mixture model. To overcome this problem, we exploit the specificity to fluorescently labeled structures of the FOB setup to identify and segment the lipid droplets. Since previous mass spectroscopy studies on adipocyte cell culture models have identified palmitoyl triacylglycerides as predominant lipid species (*Gouw and Vlugter, 1966*; *Liaw et al., 2016*), we use an absolute density value of 0.8932 g/ml for calculating the longitudinal moduli of the lipid droplets.

Here, we observed Simpson–Golabi–Behmel syndrome (SGBS) adipocytes (*Wabitsch et al., 2001*) ($N = 3$) whose nucleus and lipid droplets were stained with Hoechst and Nile red, respectively, on day 11 of adipogenic differentiation. The lipid droplets were clearly visible in the fluorescence intensity (*Figure 5a*) and showed a high mean RI value of 1.409 ± 0.004 (*Figure 5c*). The Brillouin shift of the lipid droplets of 8.25 ± 0.02 GHz was also statistically significantly higher than the Brillouin shift of the surrounding cytoplasm of 7.81 ± 0.02 GHz (*Figure 5d*). Hence, one could expect that the longitudinal modulus shows a similar trend as the Brillouin shift as it does for samples described by the two-substance mixture model. However, the longitudinal modulus of the lipid droplets (2.161 ± 0.005 GPa) was lower than that of the cytoplasm (2.398 ± 0.009 GPa) when the measured RI and extracted absolute density distributions were considered (*Figure 5e*). The longitudinal modulus of lipid droplets being lower than that of cytoplasm was consistent with previous measurement data of the speed of sound of triacylglycerides that is lower than that of water (*Gouw and Vlugter, 2006*). In order to demonstrate the effect of the RI and absolute density on the calculation of the longitudinal modulus, we calculated the longitudinal modulus under the assumption of a homogeneous RI (1.337) and absolute density (1 g/ml) distribution instead of the values measured, as it would likely be done for a stand-alone Brillouin microscope. The longitudinal modulus of lipid droplets without considering the RI and absolute densities measured results to 2.717 ± 0.022 GPa, which was 26% higher than the correctly calculated longitudinal modulus (*Figure 5f*). Our finding clearly demonstrates that the local distribution of RI and absolute density can contribute considerably to the extraction of the longitudinal modulus of the samples, especially for compartments that cannot be described by the water-based two-substance mixture model.

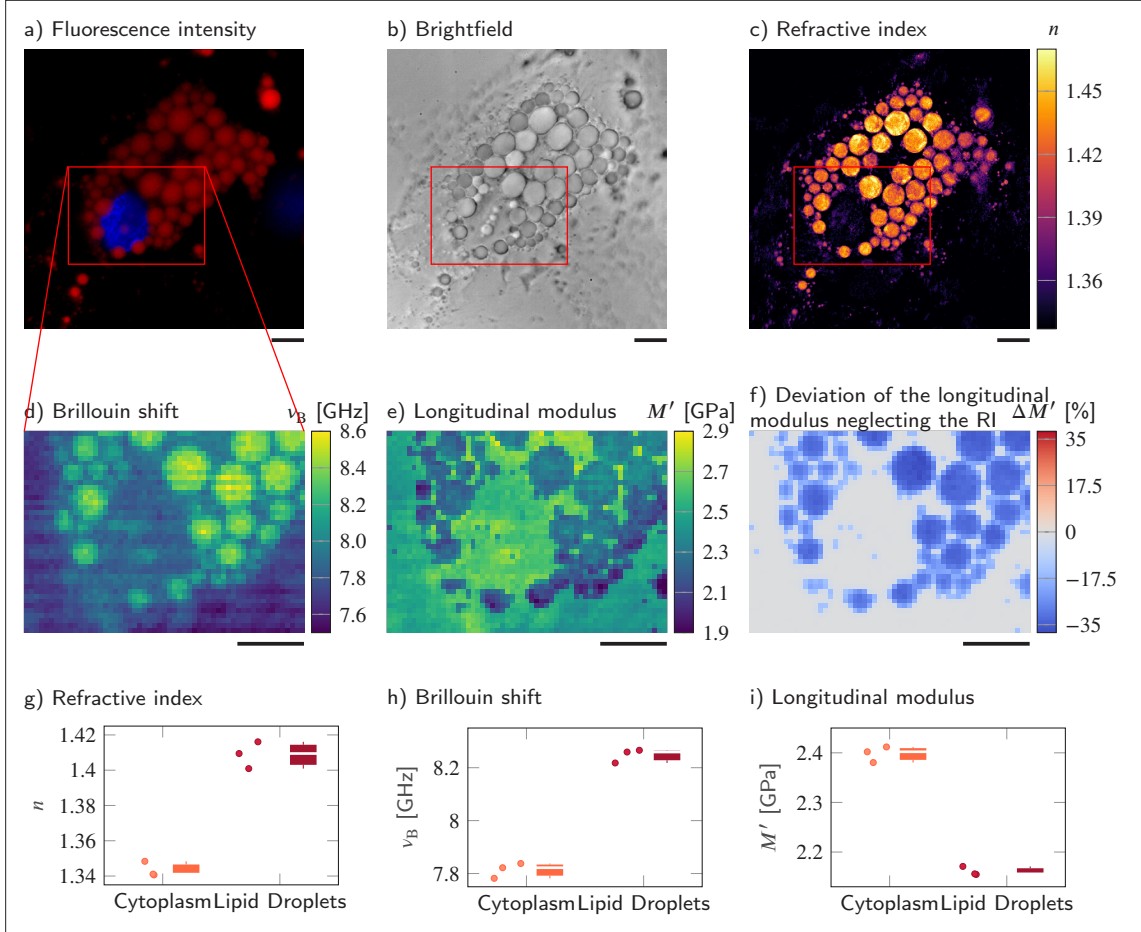

**Figure 5.** Despite a higher refractive index (RI) and Brillouin shift, the longitudinal modulus of lipid droplets is lower than that of the surrounding cytoplasm. (**a–d**) Representative maps of the (**a**) epifluorescence intensities, (**b**) brightfield intensities, (**c**) RIs, and (**d**) Brillouin shifts of an adipocyte cell. The nucleus is stained with Hoechst (blue in **a**) and lipid droplets are stained with Nile red (red in **a**). (**e**) Longitudinal modulus calculated from the RIs, absolute densities, and Brillouin shifts. (**f**) Deviation of the longitudinal modulus calculated with a homogeneous RI and absolute density value when compared to the precise longitudinal modulus in (**e**). Quantitative analysis of (**g**) the RI, (**h**) the Brillouin shift, and (**i**) the calculated longitudinal modulus taking into account the Brillouin shifts, refractive indices, and absolute densities of $N = 3$ adipocytes. The size of the Brillouin map is $57 \times 41$ pixel, resulting in an acquisition duration of 40 min. Scale bars 10 μm.

## Discussion

In this report, we experimentally demonstrated a combined epifluorescence, ODT, and Brillouin (FOB) microscopy setup. The colocalized measurements and the subsequent image analysis of the epifluorescence intensities and the RI distributions acquired by the FOB setup allowed to identify regions of different material or molecular composition. This enabled us to extract the correct absolute density from either the RI measured by applying the two-substance mixture model or from the literature in case the two-substance mixture model is not applicable. In combination with the Brillouin shift distributions measured, it was possible to accurately calculate the longitudinal moduli of a specimen. While in principle similar measurements would be possible with two separate setups individually acquiring Brillouin shift and RI, the combined setup simplifies sample handling, eliminates the necessity to locate the same cell or sample region on different setups, and substantially reduces the time between the acquisition of the different modalities from multiple minutes to a few seconds. The last point is especially important for the analysis of dynamic processes such as the formation of SGs, which otherwise would not be captured adequately. We demonstrated the working principle of the setup using an artificial cell phantom consisting of a PDMS bead embedded in a PAA bead, for which the acquired longitudinal moduli values are consistent with previous studies only when we consider the RI and the absolute density of the PDMS and PAA bead.

The setup was also applied to investigate the physical and mechanical properties of intracellular compartments in HeLa cells including nucleoplasm, cytoplasm, and nucleoli. We found that the nucleoplasm has a lower RI and absolute density than the cytoplasm while showing a higher Brillouin shift and longitudinal modulus. We further measured a statistically significantly higher peak linewidth, that is, viscosity in the nucleoli compared to the other compartments. The nucleus is described as a 'network of chromatin and other intranuclear components surrounded by a cytosol fluid' by *Wachsmuth et al., 2000*. This chromatin network was suggested to be responsible for the nucleus' mechanical response in the GHz frequency range as tested with Brillouin microscopy (*Zouani et al., 2014*), and it was shown that the mass density in the cytoskeleton network is higher than in the chromatin network although the Brillouin shift behaves in the opposite way (*Liu et al., 2019*). Hence, it seems reasonable to assume that the chromatin network in the nucleoplasm leads to the higher longitudinal modulus observed in this compartment compared to the cytoplasm, even though the mass density in the nucleoplasm is lower than in the cytoplasm. Further analysis, for example, testing the response of the longitudinal modulus to chromatin (de)condensation, can be performed to further consolidate this idea. Furthermore, nucleoli, which are formed by LLPS in the nucleoplasm, and polyQ aggregates, which undergo a rapid liquid-to-solid transition in the cytoplasm, exhibit a statistically significantly higher RI and longitudinal modulus than either nucleoplasm or cytoplasm. However, SGs in P525L HeLa cells, which are also formed by LLPS, did not show statistically significant differences in terms of RI or longitudinal modulus compared to the surrounding cytoplasm in living cells, but showed a higher longitudinal modulus compared to the cytoplasm after chemical fixation. Hence, it seems that not every condensation process is accompanied by changes of the RI, absolute density, or longitudinal modulus. Further investigation is required to reveal the underlying mechanism of how nucleoli consisting of proteins and nucleic acids maintain a higher density and longitudinal modulus than the surrounding nucleoplasm despite the dynamic behavior of compartments formed by LLPS (*Caragine et al., 2019*).

Currently, there is a vivid debate whether the Brillouin shift mainly depends on the water content of the specimen, not on its mechanical properties (*Wu et al., 2017*; *Wu et al., 2018b*; *Scarcelli and Yun, 2018*; *Bailey et al., 2019*). If we followed the idea that the water content dominates the Brillouin shift, samples with a higher water content would exhibit a lower Brillouin shift. As the RI of the cytoplasm and the nucleoplasm of the HeLa cells measured here is linearly proportional to the mass density of macromolecules in water solution (*Barer, 1952*; *Popescu et al., 2008*) and the refraction increments of both compartments are similar (*Zhao et al., 2011*; *Zangle and Teitell, 2014*), the lower RI of the nucleoplasm compared to the cytoplasm indicates that the nucleoplasm has a higher water content than the cytoplasm. However, the nucleoplasm exhibits a higher Brillouin shift and longitudinal modulus than the surrounding cytoplasm. Hence, this result indicates that the Brillouin shift and the longitudinal modulus are not only governed by the water content, but are at least substantially influenced by the mechanical properties of the specimen.

An important aspect of the calculation of the longitudinal moduli is the extraction of the densities of the samples. For samples or compartments that can be described by the two-substance mixture model, we exploited the linear relation between the RI and the mass density to calculate the absolute density value (*Barer, 1952*; *Popescu et al., 2008*; *Zangle and Teitell, 2014*). However, as the partial specific volume of the dry fraction is unknown, this approach might overestimate the absolute density by approximately 10% (see Materials and methods). We find that in all samples measured here where the two-substance mixture model can be applied, neglecting the contribution of RI and density to the longitudinal modulus still yields a similar tendency for the longitudinal modulus and Brillouin shift (i.e., a high Brillouin shift means a high longitudinal modulus and vice versa), but doing so might lead to a systematic error for the longitudinal modulus. For cell compartments mainly containing a single substance, where this model cannot be applied, for example, lipid droplets in adipocytes, we used the specificity provided by the epifluorescence imaging to identify the respective regions and employed the literature value for the absolute density in this region. Using this approach, we found that although the RI and Brillouin shift of the lipid compartments in adipocytes are higher than those values of the cytoplasm, the resulting longitudinal modulus is actually lower when taking into account the RI and absolute density distribution. This illustrates that RI and absolute density do not cancel out for every cell and compartment – an implicit assumption in many studies acquiring only the Brillouin shift – and that RI and absolute density have to be known in order to precisely calculate the longitudinal modulus.

However, both the calculation of the absolute density from the RI and the identification of regions not described by the two-substance mixture model rely on the knowledge of the molecular composition of the sample. In order to calculate the absolute density from the RI, the refraction increment has to be known, which, albeit comparable for proteins and nucleic acids, might slightly vary between different cell compartments depending on their composition. Obviously, the composition also plays an important role when selecting the correct literature value for the absolute density of compartments where the two-substance mixture model is not applicable. As the molecular composition cannot be resolved exactly by the FOB microscope, we used the refraction increment or absolute density of the constituent likely predominant in a certain compartment. This might lead to a slight deviation of the absolute density from the exact value, for example, in the membrane-rich perinuclear region of HeLa cells where the absolute density might be underestimated. To overcome this issue and use the appropriate refraction increment or absolute density for a mixture of different proteins, nucleic acids, or phospholipids, more sophisticated labeling and staining of different molecules and the use of several fluorescence channels might allow identifying multiple substances. Also, the absolute concentration of different molecules could be directly measured from the intensity of Raman scattering signals (*Oh et al., 2019*), an imaging extension that could be added for future studies to the FOB setup presented here (*Traverso et al., 2015*; *Mattana et al., 2018*).

Further improvements of the setup could include moving to a laser source with a wavelength of 660 nm or longer to reduce cell damage due to phototoxicity (*Nikolić and Scarcelli, 2019*). This would allow a higher laser power at the sample plane for Brillouin microscopy, which reduces the acquisition time and could help analyzing dynamic processes. To correlate the RI value and Brillouin frequency shift of the samples at the same wavelength, the FOB setup uses the same laser source for ODT and Brillouin mode. Brillouin spectroscopy requires a laser with an extremely narrow linewidth that, hence, has a high temporal coherence length. While the coherent nature of the laser illumination makes ODT susceptible to speckle noise, the ODT system used achieves an RI uncertainty of $4.15 \times 10^{-5}$ (which corresponds to a difference in protein concentration of 0.22 mg/ml) (*Kim and Guck, 2020*) that is sufficient to pick up the RI differences between the various regions of the cells characterized here. The speckle noise could be further reduced by using a dynamic diffuser (*Choi et al., 2011*) or numerical filtering approaches (*Bianco et al., 2018*). Furthermore, the setup could be enhanced to measure not only longitudinal phonons, but also transverse phonons, which are related to the shear modulus and can generally not propagate in liquids. Hence, this could help to discriminate liquid-like versus solid-like materials (*Kim et al., 2016b*; *Prevedel et al., 2019*). The limitation of ODT to weakly scattering samples like single cells or beads could be overcome by the implementation of tomogram reconstruction algorithms taking into account multiple light scattering in the sample (*Lim et al., 2019*; *Chowdhury et al., 2019*). This would enable the setup to measure tissues and organisms.

Although Brillouin microscopy was introduced to biology more than a decade ago (*Scarcelli and Yun, 2008*), its relevance to biological questions is sometimes still viewed skeptically in the field of mechanobiology. This is mainly due to the fact that the main quantity measured – the longitudinal modulus – relates to the rarely acquired compressibility in a frequency range to which cells might not be sensitive. However, multiple studies observed changes of the longitudinal modulus due to biophysical processes. The modulus changes due to the inhibition of actin polymerization (*Scarcelli et al., 2015*; *Antonacci and Braakman, 2016*) and after spinal cord injury (*Schlüßler et al., 2018*), after UV-induced polymer crosslinking (*Scarcelli and Yun, 2008*), or actin polymerization into a gel (*Scarcelli et al., 2015*). For various samples, for example, cells under osmotic shock, bovine lenses, or zebrafish tissue, phenomenological correlations with the Young's modulus have been found (*Scarcelli and Yun, 2011*; *Scarcelli et al., 2015*; *Schlüßler et al., 2018*), implying that for these tissues the two moduli could serve as a proxy for each other. Furthermore, Brillouin microscopy gives access to other quantities besides the compressibility. The viscosity and shear modulus of a sample are also accessible by Brillouin microscopy by evaluating the peak linewidth and observing transverse phonons (*Antonacci et al., 2018*; *Kim et al., 2016b*; *Prevedel et al., 2019*) and can be influenced, for example, by liquid-to-solid phase transitions. For the observed, physically real variations of the longitudinal modulus and viscosity to affect biophysical processes, it is moreover not necessary that cells or organisms are able to sense these mechanical differences directly in the first place, as would be relevant in the context of mechanosensing. It seems entirely possible that the variations in the longitudinal modulus at GHz frequencies detected in cells will turn out to be reporting on local changes in

intermolecular interactions, water mobility and hydration shells, and other aspects relevant for emergent, supramolecular processes, which are important in their own right. We thus believe that Brillouin microscopy can contribute to open questions in biology, but further studies are necessary to finally establish its relevance.

In conclusion, the FOB setup allows a precise calculation of the longitudinal modulus from the measured RI and Brillouin shift even for samples with a heterogeneous RI and absolute density distribution. This enables quantitative measurements of the mechanical properties of single cells and their compartments and might potentially contribute to a better understanding of physiological and pathological processes such as phase separation and transition in cells as a response to external stress.

## Materials and methods
### Optical setup

The FOB microscope setup combines ODT, Brillouin microscopy, and epifluorescence imaging in the same optical system. It allows to obtain quantitative maps of the RIs, the Brillouin shifts, and the fluorescence and brightfield intensities of a sample.

In order to acquire the 3D RI distribution, ODT employing Mach–Zehnder interferometry was applied. Besides small modifications necessary for the combination with Brillouin microscopy, the ODT part of the setup is identical to the one presented in *Abuhattum et al., 2018*. As laser source, a frequency-doubled Nd-YAG laser (Torus 532, Laser Quantum Ltd, UK) with a wavelength of 532 nm and a maximum output power of 750 mW is used for both ODT and Brillouin microscopy. The laser was chosen as it offers a very low amplified spontaneous emission intensity of 110 dB necessary for Brillouin measurements. The main beam of the laser is coupled into a single-mode fiber and split into two beams by a $2 \times 2$ fiber coupler. One beam is used as the reference for the Mach–Zehnder interferometer. The other beam is collimated and demagnified through a tube lens with a focal length of 175 mm and a ×40/1.0 NA water dipping objective lens (Carl Zeiss AG, Germany) and illuminates the sample in a custom-built inverted microscope. To allow to reconstruct a 3D RI tomogram of the sample, the sample is illuminated under 150 different incident angles. The illumination angles are generated by a dual-axis galvanometer mirror (GVS012/M, Thorlabs Inc, USA), which is placed at the conjugate plane of the sample and diffracts the illumination beam. The diffracted beam is collected by a ×63/1.2 NA water immersion objective lens (Carl Zeiss AG) and a tube lens with a focal length of 200 mm. The sample and the reference beam then interfere at the image plane of a CCD camera (FL3-U3-13Y3M-C, FLIR Systems Inc, USA), which records the generated spatially modulated hologram of the sample. In some cases, the hologram additionally shows parasitic interference patterns originating from the protective window in front of the CCD camera (e.g., *Figure 2b*; this is a general limitation of the ODT setup due to the coherent nature of the laser source). The setup achieves a spatial resolution of 0.25 μm within the lateral plane and 0.5 μm in the axial direction.

In order to switch to Brillouin microscopy mode, a motorized mirror is moved into the beam path guiding the light towards an additional lens with a focal length of 300 mm. In combination with the upper tube lens, this ensures a collimated beam before the microscope objective and effectively creates a laser focus at the sample plane. Hence, in Brillouin mode the galvanometer mirrors are located at the Fourier conjugate plane of the sample and can move the laser focus in the sample plane (*Figure 1a*, inset). This allows to scan the laser focus over the sample by adjusting the galvanometer voltage. The relation between the applied galvanometer voltage and the resulting focus position is calibrated by acquiring images of the laser foci with the ODT camera and extracting the foci positions for different galvanometer voltages. The Brillouin scattered light is collected in the backscattering configuration with the same objective used for ODT and coupled into a single-mode fiber that acts as a pinhole confocal to the illumination fiber and delivers the light to a two-stage VIPA Brillouin spectrometer (*Scarcelli and Yun, 2011*; *Scarcelli et al., 2015*). This results in a spatial resolution of 0.4 μm within the lateral plane and approximately 1 μm in the axial direction. In the spectrometer, the beam is collimated and passes through the iodine absorption cell, which blocks the Rayleigh scattered and reflected light. The beam is then guided to two VIPA interferometers (OP-6721-3371-2, Light Machinery, Canada) with 30 GHz free spectral range and a spectral resolution of approximately $\delta\nu = 350\,\mathrm{MHz}$, which is comparable to values reported for other VIPA-based setups (*Antonacci et al., 2013*) but lower than the spectral resolution achievable with stimulated Brillouin scattering setups of

around 100 MHz (*Remer et al., 2020*). The Brillouin spectrum is imaged with an sCMOS camera (Neo 5.5, Andor, USA) with a typical exposure time of 0.5 s at a laser power of 10 mW at the sample.

Furthermore, the laser frequency is stabilized to the absorption maximum of a transition line of molecular iodine by controlling the laser cavity temperature. This allows to attenuate the intensity of the Rayleigh scattered light entering the Brillouin spectrometer, eliminates potential laser frequency drifts (*Meng et al., 2014*; *Schlüßler et al., 2018*), and simplifies the mechanical alignment of the spectrometer as no masks for blocking the elastically scattered light are necessary. To generate an error signal for the frequency stabilization loop, a small fraction of the laser light is frequency shifted by 350 MHz by an acousto-optic modulator (AOM 3350-125, EQ Photonics GmbH, Germany) and guided through an absorption cell (TG-ABI-Q, Precision Glass Blowing, USA) filled with iodine $I_2$. The beam intensity is measured before and after the absorption cell by two photodetectors (PDA36A2, Thorlabs Inc) and a data acquisition card (PicoScope 2205A, Pico Technology, UK). The quotient of both intensities is a measure for the absorption due to the iodine vapor. The laser cavity temperature is then controlled with a custom C++ software LQTControl to achieve an absorption of 50% for the frequency-shifted stabilization beam, which leads to maximum absorption for the not-shifted main beam.

To realize epifluorescence imaging, an incoherent beam from a white light halogen lamp (DC-950, Dolan-Jenner Industries Inc, USA) is coupled into the setup by a three-channel dichroic mirror (FF409/493/596-Di01−25 × 36, Semrock, USA). The bandwidth of the excitation and emission beam is selected by two motorized filter sliders equipped with band-pass filters in front of the halogen lamp and the CCD camera. A white light LED (Thorlabs, USA) coupled into the Brillouin illumination path allows to observe a brightfield image of the sample during Brillouin acquisition. Since fluorescence imaging and ODT use the same objective, the acquired fluorescence images are guaranteed to focus the central plane of the acquired RI tomogram.

The two cameras and all moveable optical devices of the setup are controlled with a custom acquisition program written in C++. The software allows to control all three imaging modalities and stores the acquired data as an HDF5 file.

## Refractive index tomogram reconstruction

From the spatially modulated holograms recorded, the complex optical field of the light scattered by the sample is retrieved by a field retrieval algorithm based on the Fourier transform (*Cuche et al., 2000*). The RI tomogram of the sample is reconstructed from the retrieved optical fields with various incident angles via the Fourier diffraction theorem. The detailed procedure for the tomogram reconstruction is presented in *Kim et al., 2014*; *Müller et al., 2015*. The field retrieval and tomogram reconstruction were performed by custom-made MATLAB (The MathWorks, Natick) scripts. From the reconstructed RI tomograms, subcellular compartments are segmented based on the RI and epi-fluorescence signals. First, cell regions are segmented from background by applying the Otsu's thresholding method, and the watershed algorithm is used to determine individual cells in the RI tomograms. Then, epifluorescence images of the fluorescence-labeled subcellular compartments (e.g., nuclei, polyQ aggregates in HeLa cells, nuclei and lipid droplets in adipocytes) are colocalized with the RI tomograms to segment the compartments. In the nuclei of the HeLa cells, the RI tomogram regions having higher RI values than the surrounding nucleoplasm are segmented by the Otsu's thresholding method and identified as nucleoli. The detailed segmentation procedure is described elsewhere (*Schürmann et al., 2016*; *Kim and Guck, 2020*), and the source code for the segmentation can be found at https://github.com/OpticalDiffractionTomography/NucleiAnalysis.

## Brillouin shift evaluation

To evaluate the Brillouin shift $\nu_B$, a custom MATLAB program is used. Details of the evaluation process can be found in *Schlüßler et al., 2018*.

## Calculation of the longitudinal modulus

The longitudinal modulus $M'$ is determined by

$$M' = \rho \left( \frac{\lambda \nu_B}{2n \cos (\Theta/2)} \right)^2 \tag{1}$$

where the wavelength $\lambda$ of the laser source and the scattering angle $\Theta$ are known parameters of the setup. The RI $n$ and the Brillouin shift $\nu_B$ of the sample are measured using the FOB microscope. The absolute density $\rho$ can be calculated for the majority of biological samples from the RI assuming a two-substance mixture. The absolute density is given by *Barer, 1952*; *Davies and WilkinsNS, 1952*; *Zangle and Teitell, 2014*; *Popescu et al., 2008*; *Schlüßler et al., 2018*

$$\rho = \frac{n - n_{\text{fluid}}}{\alpha} + \rho_{\text{fluid}} \cdot \left(1 - \rho_{\text{dry}} \cdot \bar{\nu}_{\text{dry}}\right). \tag{2}$$

with the RI $n_{\text{fluid}}$ of the medium, the refraction increment $\alpha$ ($\alpha = 0.190 \, \text{mL/g}$ for proteins and nucleic acid [*Zhao et al., 2011*; *Zangle and Teitell, 2014*; *Biswas et al., 2021*]), the absolute density $\rho_{\text{fluid}}$ of the medium, the absolute density $\rho_{\text{dry}}$, and the partial specific volume $\bar{\nu}_{\text{dry}}$ of the dry fraction. In case of $\rho_{\text{dry}} \ll \frac{1}{\bar{\nu}_{\text{dry}}}$, this can be simplified to

$$\rho \approx \frac{n - n_{\text{fluid}}}{\alpha} + \rho_{\text{fluid}}. \tag{3}$$

This simplification leads to an overestimation of the absolute density and, hence, the longitudinal modulus, of around 10% for , for example, HeLa cells, which we believe to be acceptable.

For certain cell types, for example, adipocyte cells, the two-substance mixture model cannot be applied for all cell compartments, that is, the lipid droplets inside these cells do only consist of lipids. Applying the two-substance model here leads to an unphysiological overestimation of the absolute density. Hence, in special cases the absolute density cannot be inferred from the RI and has to be estimated from the literature. This is possible with the FOB microscope since fluorescence labeling of the lipid droplets allows to identify cell regions governed by, for example, lipids.

In order to calculate the longitudinal modulus and visualize the measurement results of the FOB microscope, a custom MATLAB program FOBVisualizer is used. The software allows to adjust the spatial alignment of the Brillouin and ODT measurements by cross-correlating the two-dimensional maps acquired by both modalities and shifting the Brillouin maps towards the highest correlation coefficient.

## Statistical analysis

For the statistical analysis of the RI and longitudinal modulus differences between cytoplasm, nucleoplasm, and nucleoli (*Figures 2–4*), the Kruskal−Wallis test in combination with a least significant difference post-hoc test was used. Asterisks indicate the significance levels: *$p<0.05$, **$p<0.01$, ***$p<0.001$, and ****$p<0.0001$. In box-and-whisker plots, the center lines indicate the medians, the edges of the boxes define the 25th and 75th percentiles, the red plus signs represent data points outside the $\pm 2.7\sigma$ range, which are considered outliers, and the whiskers extend to the most extreme data value that is not an outlier.

## Cell phantom preparation

Artificial cell phantoms, consisting of PDMS (Dow Corning Sylgard 184) particles embedded in larger PAA microgel beads, were produced as follows. The PDMS particles were generated by vortex-mixing a solution of 1 g PDMS (10:1 w/w, base/curing agent) dispersed in 10 ml of 2% w/v poly(ethylene glycol) monooleate (Merck Chemicals GmbH, Germany) aqueous solution. After mixing, the emulsion was kept overnight in an oven at 75°C to allow the polymerization of the PDMS droplets. The size dispersion of the PDMS particle was reduced by centrifugation and removing all particles with a diameter larger than 5 μm. The final solution, containing PDMS particles with a diameter lower than 5 μm, was washed three times in Tris-buffer (pH 7.48) and resuspended in 1% w/v Pluronic F-127 (Merck Chemicals GmbH) Tris-buffer.

1 μl of concentrated PDMS particles were added to 1001 PAA pre-gel mixture with a total monomer concentration of 11.8% w/v. This solution was used as a dispersed phase in a flow-focusing microfluidic device to produce PAAm microgel beads, as previously described in *Girardo et al., 2018*, embedding PDMS particles. N-hydroxysuccinimide ester (0.1% w/v, Merck Chemicals GmbH) was added to the oil solution to functionalize the phantoms with Alexa 488. Precisely, 100 μl of Alexa Fluor hydrazide 488 (Thermo Fisher Scientific, Germany) in deionized water (1 mg/ml) was added to 100 μl phantom pellet and incubated overnight at 4°C. The unbonded fluorophores were removed by three washings in PBS. The final functionalized phantoms were stored in PBS at 4°C.

## Cell preparation

The stable HeLa cell line expressing GFP fused to the N terminus of NIFK (nucleolar protein interacting with the FHA domain of MKI67) was kindly provided by the lab of Anthony Hyman (Max Planck Institute of Molecular Cell Biology and Genetics). The cells were cultured in Dulbecco's modified Eagle's medium (DMEM) (31966-021, Thermo Fisher), high glucose with GlutaMax medium (61965-026, Gibco) under standard conditions at 37°C and 5% $CO_2$. The culture medium was supplemented with 10% fetal bovine serum (FBS) and 1% penicillin-streptomycin. The cells were subcultured in a glass-bottom Petri dish (FluoroDish, World Precision Instruments Germany GmbH) 1 day prior to the measurement, and the culture medium was exchanged to Leibovitz's L-15 medium without phenol red (21083027, Thermo Fisher Scientific) prior to imaging. For staining nuclei, the cells were stained with Hoechst (1 dilution) for 10 min and washed with fresh Leibovitz's L-15 medium prior to imaging.

The wild-type HeLa cells transiently expressing amyloid (Q103-GFP) aggregates were cultured in DMEM (31966-021, Thermo Fisher), high glucose with GlutaMax medium (61965-026, Gibco) under standard conditions at 37°C and 5% $CO_2$. The culture medium was supplemented with 10% FBS and 1% penicillin-streptomycin. The cells were subcultured in a glass-bottom Petri dish (FluoroDish, World Precision Instruments Germany GmbH) 2 days prior to the measurement. One day prior to the measurement, the cells were transiently transfected with pcDNA3.1-Q103-GFP using Lipofectamine 2000 (Invitrogen, Carlsbad, CA). Directly before the imaging, the culture medium was exchanged to Leibovitz's L-15 medium without phenol red (21083027, Thermo Fisher Scientific).

The HeLa cells FUS-GFP WT (wild-type) and FUS-GFP[P525L] (disease model for ALS) were kindly provided by the lab of Anthony Hyman (Max Planck Institute of Molecular Cell Biology and Genetics). The cells were cultured in 89% DMEM supplemented with 10% FBS (Sigma-Aldrich; F7524) and 1% penicillin-streptomycin under standard conditions at 37°Cand 5% $CO_2$. One day before the experiment, the cells were transferred to a 35 mm glass-bottom Petri dish (FluoroDish, World Precision Instruments Germany GmbH). 30 min prior to the measurements, the culture medium was exchanged to Leibovitz's L-15 medium without phenol red (21083027, Thermo Fisher Scientific) and the noncontrol samples were treated with 5 mM sodium arsenite. For fixation, the arsenite-treated cells were washed with PBS, fixed with 4% paraformaldehyde for 10 min at room temperature, washed with PBS twice, and left in PBS for FOB microscopy measurements.

## Adipocyte preparation

SGBS preadipocytes were cultured and differentiated as described previously (*Wabitsch et al., 2001*; *Fischer-Posovszky et al., 2008*). For regular cell culture, cells were maintained in DMEM/nutrient F-12 Ham (Thermo Fisher) supplemented with 4 µM panthotenic, 8 µM biotin (Pan/Bio), 100 U/ml penicillin/100 µg/ml streptomycin (=OF-medium) with 10% FBS (OF-medium + FBS, Thermo Fisher) at 37°C in T75 flasks. For adipogenic differentiation, cells were washed with PBS, detached using TrypLE Express (Thermo Fisher), and seeded onto glass-bottom Petri dishes (FluoroDish, World Precision Instruments Germany GmbH, 35 mm, $10^5$ cells). After 24 hr, cells were washed three times with serum-free OF-medium, and differentiation medium was added, consisting of OF-medium complemented with 10 µg/ml human transferrin (Sigma-Aldrich), 20 nM human insulin (Sigma-Aldrich), 2 µM rosiglitazone (Cayman), 100 nM dexamethasone (Sigma-Aldrich), 250 µM 3-isobutyl-1-methylxantine IBMX (Sigma-Aldrich), 100 nM cortisol (Sigma-Aldrich), and 0.2 nM triiodothyronine T3 (Sigma-Aldrich). On day 4, the medium was exchanged to OF-medium supplemented with only transferrin, insulin, cortisol, T3 (concentrations as above). The medium was replaced every fourth day. Cells were probed on day 11 of adipogenic differentiation.

## Cell lines

HeLa WT cells were a kind gift of F. Buchholz (Technische Universitat Dresden, Germany). The HeLa cell line transfected with GFP:NIFK, HeLa WT FUS-GFP, and HeLa P525L FUS-GFP was a kind gift of A. Hyman (Max Planck Institute of Molecular Cell Biology and Genetics, Dresden, Germany). SGBS adipocytes were a kind gift of Prof. Martin Wabitsch (Centre for Hormonal Disorders in Children and Adolescents – Ulm University Hospital).

No authentication was performed. All cell lines tested negative for mycoplasma contamination. No commonly misidentified cell lines were used.

## Code availability

The source code of LQTControl, the program to stabilize the laser cavity temperature, is open source and can be found on GitHub (https://github.com/BrillouinMicroscopy/LQTControl; *Schlüßler, 2018*). The same is true for BrillouinAcquisition, the program for controlling and data acquisition of the FOB microscope (https://github.com/BrillouinMicroscopy/BrillouinAcquisition; *Schlüßler, 2017*), BrillouinEvaluation, used for evaluating Brillouin data (https://github.com/BrillouinMicroscopy/BrillouinEvaluation; *Schlüßler, 2016*), and FOBVisualizer, used for viewing FOB microscopy data (https://github.com/BrillouinMicroscopy/FOBVisualizer ; *Schlüßler, 2019*). The MATLAB scripts for cell segmentation and ODT reconstruction can be found under https://github.com/OpticalDiffractionTomography/Nuclei-Analysis (*Kim, 2022a* copy archived at swh:1:rev:ee5da0592cb893bac393f5cd19c223a518495c4a) and https://github.com/OpticalDiffractionTomography/ODT_Reconstruction (*Kim, 2022b* copy archived at swh:1:rev:e9a63082bb55564fc3edbcf48108e1e4b6f12458), respectively.

## Acknowledgements

We thank Anthony Hyman from the Max Planck Institute of Molecular Cell Biology and Genetics for providing the stable HeLa cell lines, Prof. Martin Wabitsch from the Centre for Hormonal Disorders in Children and Adolescents – Ulm University Hospital for providing the SGBS preadipocytes, and the Center for Molecular and Cellular Bioengineering Light Microscopy Facility (partly funded by the State of Saxony and the European Fund for Regional Development – EFRE) for technical support. Financial support from the Deutsche Forschungsgemeinschaft (SPP 2191-Molecular mechanisms of functional phase separation, grant agreement number 419138906 to SAl and JG), Volkswagen Stiftung (grant agreement number 92847 to SAl and JG), the Alexander von Humboldt-Stiftung (Alexander von Humboldt-Professorship to JG) are gratefully acknowledged. AH is supported by the NOMIS foundation and the Hermann und Lilly Schilling-Stiftung fur medizinische Forschung im Stifterverband.

## Additional information

### Funding

| Funder | Grant reference number | Author |
|---|---|---|
| Deutsche Forschungsgemeinschaft | 419138906 | Simon Alberti Jochen Guck |
| Volkswagen Foundation | 92847 | Simon Alberti Jochen Guck |
| Alexander von Humboldt-Stiftung | | Jochen Guck |
| NOMIS Stiftung | | Andreas Hermann |
| Hermann und Lilly Schilling-Stiftung | | Andreas Hermann |

The funders had no role in study design, data collection and interpretation, or the decision to submit the work for publication.

### Author contributions

Raimund Schlüßler, Conceptualization, Data curation, Formal analysis, Funding acquisition, Investigation, Methodology, Project administration, Resources, Software, Validation, Visualization, Writing – original draft, Writing – review and editing; Kyoohyun Kim, Conceptualization, Data curation, Formal analysis, Funding acquisition, Investigation, Methodology, Resources, Software, Validation, Visualization, Writing – original draft, Writing – review and editing; Martin Nötzel, Shada Abuhattum, Timon Beck, Formal analysis, Investigation, Writing – review and editing; Anna Taubenberger, Conceptualization, Funding acquisition, Investigation, Methodology, Supervision, Writing – review and editing; Paul Müller, Formal analysis, Software, Writing – review and editing; Shovamaye Maharana, Gheorghe Cojoc, Investigation, Writing – review and editing; Salvatore Girardo, Funding acquisition, Methodology, Supervision, Writing – review and editing; Andreas Hermann, Conceptualization, Funding

acquisition, Supervision, Writing – review and editing; Simon Alberti, Conceptualization, Funding acquisition, Project administration, Supervision, Writing – review and editing; Jochen Guck, Conceptualization, Funding acquisition, Methodology, Project administration, Supervision, Writing – review and editing

#### Author ORCIDs
Raimund Schlüßler ![ORCID] http://orcid.org/0000-0003-3752-2382
Kyoohyun Kim ![ORCID] http://orcid.org/0000-0003-1808-775X
Martin Nötzel ![ORCID] http://orcid.org/0000-0002-6442-9899
Andreas Hermann ![ORCID] http://orcid.org/0000-0002-7364-7791
Simon Alberti ![ORCID] http://orcid.org/0000-0003-4017-6505

#### Decision letter and Author response
Decision letter https://doi.org/10.7554/eLife.68490.sa1
Author response https://doi.org/10.7554/eLife.68490.sa2

## Additional files

#### Supplementary files
• Supplementary file 1. Average values and standard errors of the mean of the refractive index (RI) $n$, Brillouin shift $\nu_B$, absolute density $\rho$, longitudinal modulus $M'$, and linewidth $\Delta_B$ for the cytoplasm (cyto), nucleoplasm (np), and nucleoli (nl) of 139 wild-type HeLa cells.

• Supplementary file 2. Kruskal–Wallis p-values when comparing the refractive index (RI) $n$, Brillouin shifts $\nu_B$, mass densities $\rho$, longitudinal moduli $M'$, and linewidths $\Delta_B$ of the cytoplasm (cyto), nucleoplasm (np), and nucleoli (nl) of 139 wild-type HeLa cells, respectively.

• Supplementary file 3. Average values and standard errors of the mean of the refractive index (RI) $n$, Brillouin shift $\nu_B$, absolute density $\rho$, and longitudinal modulus $M'$ for the cytoplasm and polyglutamine (polyQ) aggregates of 22 wild-type HeLa cells.

• Supplementary file 4. Average values and standard errors of the mean of the refractive index (RI) $n$ and longitudinal modulus $M'$ for different conditions and compartments of P525L HeLa cells.

• Transparent reporting form

#### Data availability
The data sets generated during and/or analyzed during the current study are available from figshare under the following link: https://doi.org/10.6084/m9.figshare.c.5347778.

The following dataset was generated:

| Author(s) | Year | Dataset title | Dataset URL | Database and Identifier |
|---|---|---|---|---|
| Raimund S, Kyoohyun K, Martin N, Anna T, Shada A, Timon B, Paul M, Shovamayee M, Gheorghe C, Salvatore G, Andreas H, Simon A, Jochen G | 2021 | Combined fluorescence, optical diffraction tomography and Brillouin microscopy | https://doi.org/10.6084/m9.figshare.c.5347778 | figshare, 10.6084/m9.figshare.c.5347778 |

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
