## [Editor Report]

This is an important and timely contribution that introduces a new approach that combines Brillouin microscopy with fluorescence (FOB) to measure the mechanical properties in terms of longitudinal moduli for viscoelastic materials in cells. This approach has many promising applications, which the authors articulate, and could be important as a new and complementary modality for investigating the mechanical properties of soft materials, specifically membrane-bound and membrane-less organelles.

---

## [Decision Letter]

**Decision letter after peer review:**

Thank you for submitting your article "Combined fluorescence, optical diffraction tomography and Brillouin microscopy" for consideration by *eLife*. Your article has been reviewed by 3 peer reviewers, and the evaluation has been overseen by a Reviewing Editor and Anna Akhmanova as the Senior Editor. The reviewers have opted to remain anonymous.

Essential revisions:

1) One of the reviewers asked for a deeper discussion of the biological relevance of the longitudinal modulus measurement at high frequencies. Are these high frequencies likely to be of importance? Perhaps in analogy with THz spectroscopy that is useful for probing the frequency dependence of dielectric responses, these high frequency longitudinal moduli might be of import for querying the elastic and / or dielectric responses. Please add a relevant discussion.

2) One of the reviewers asks for a deeper, technical discussion of the choice of the diffraction tomography method, specifically the off-axis holography with laser illumination approach. Are there issues of phase instability and speckles and how are these alleviated? Addition of data in the SI to address this point would be very helpful.

3) Please add a discussion to address the concern of independent calibration methods raised by reviewer 3. The issue is two-fold: How well does the method work with samples / condensates that are of lower density? The reconstituted G3BP1/2 based stress granule system, already studied in the Alberti lab, would be a good target because the protein is only 7-8 fold more concentrated in the stress granule facsimiles. Additionally, the addition of FRAP data as a comparative standard would be useful.

4) In addition, please make revisions to address the series of points raised by reviewer 1. And please make these changes in a way that is readily visible.

*Reviewer #1 (Recommendations for the authors):*

In this interesting study by Schlüßler et al., the authors combined Brillouin microscopy with ODT and epi-fluorescence imaging and applied the same to study physical properties of biological materials including nucleoplasm, cytoplasm, phase-separated organelles, and adipocytes. The results are largely convincing and offer interesting insights into the material properties of these materials. There is a critical need for new methodologies to study the physical properties of biomolecular condensates in living cells under normal and pathological conditions. Hence, the current study can be significant. However, I have noted several points after carefully reading the manuscript, which is described below.

1. Abstract: "Investigating the cell nucleus, we find that it has lower density but higher longitudinal modulus" compared to what exactly?

2. Abstract: The authors state that they have measured "absolute density with molecular specificity". Is this true? The discussion and conclusion section (p14, lines # 417-427) nicely summarizes the potential limitations of the current approach and offers future strategies on the same issue to improve upon the current foundation.

3. P2. Line #55 – 63. The authors build their case of employing a new integrative approach (FOB microscopy) for correct longitudinal modulus measurements, however, it is not quite clear how simplifications such as using measured mass density with a priori knowledge of RI, or obtaining RI values from separate measurements lead to erroneous results of longitudinal modulus.

4. In my opinion, the description of the results obtained in this study by the application of FOB microscopy as stated at the end of the introduction section is rather qualitative and can be improved.

5. Images of RI in Figure 2b, 3b, 4b and d have concentric rings throughout the images. What are these rings? If the RI is calculated from the image, then are these rings artifacts of image processing? Please explain.

6. Page 7. The authors state that the Brillouin shift of the nucleoplasm is "significantly" different than the cytoplasm. I have a question regarding the use of the word "significantly" here. The authors show a moduli difference of 0.038 GPa. While this may be significant in the absolute sense, it's may not be a huge difference relative to the measured moduli of the two materials. The same thing could be said for the refractive index differences, but in that case, the range of known refractive indices is small. So, please explain.

7. The authors show that the RI and the density of the cytoplasm and nucleoplasm are comparable to water. I am curious to see what value of the longitudinal modulus will they find if they measure the same for pure water samples. For the correct modulus to be obtained, wouldn't the Brillouin shift be very different from that of the cytoplasm and nucleoplasm? In such a case, is it reasonable to assume water values for the density and the refractive index and measure only the Brillouin shift to obtain the moduli?

8. The authors state that "These findings imply that membrane-less compartments formed by phase separation……in spite of the thermodynamic instability inherent in this state" This information is nice and needed. But the authors need to explain why one should care about this longitudinal modulus, what does it represent, and how does it affect the material properties? In other words, the meaning of this modulus and its relation to the mechanical behavior of these materials is not clear. A discussion is enough to address this point.

9. On the previous point: The authors need to comment on these extremely large longitudinal moduli, as reported for polyQ aggregates and others. Previous studies showed that the elastic modulus of the ECM is about 1 kPa (additionally see table 1 in this report: https://www.ncbi.nlm.nih.gov/pmc/articles/PMC4553184/), so what does a value of 1 GPa for the longitudinal modulus mean? a good discussion of the meaning of the longitudinal modulus and its relation to the material properties and other moduli will be helpful.

10. Page 10, lines 265-268; I see that "significantly" is used here to describe statistical significance, and maybe it's used in the same sense in the other cases too. However, the authors need to specify as it can be easily misunderstood as a significant difference in magnitude and not in statistical similarity.

11. Page 10, lines 270-272; I am having trouble in understanding why the authors are using the standard error of the mean as opposed to the standard deviation as a measure of uncertainty. This gives an impression that the technique has superior accuracy. the variability shown in the plots is much larger than these errors (orders of magnitudes). Is this variability coming from the cell-to-cell and condensate to condensate variation or is it coming from the measurement itself? This data is not shown for the phantom cell controls (please include it in the SI). If this variability is present in that system as well, the authors may try a different system for the control to assess the inherent error of the technique, such as an aqueous two-phase system (e.g. peg and dextran). These systems are also closer to the intracellular fluid, so they may serve as better control. This would help the readers assess the accuracy of the technique itself. Alternatively, the authors should give a convincing argument for why are the SEMs a better measure of the variability of the measurement (or the uncertainty)? It's also crucial to understand the source of the variability in the refractive indices and moduli. If it is coming from the instrument, then SEMs should not be used. If it is coming from the cell-to-cell variation, then the authors may need to show data on a control system that does not have variations so that the readers can understand the level of accuracy of these measurements. If it is indeed from cell-to-cell variation and not from the experimental technique, then the use of SEMs is reasonable.

12. Are the errors in the case of FUS and polyQ condensates estimated from different condensates in the same cell or from one condensate in multiple cells? this is important to judge the accuracy of this measurement and the possible effect of reducing the ROI, would reducing the ROI of the Brillouin experiment on polyQ aggregates give the same result?

13. Regarding the difference in the RI and longitudinal moduli of the nucleoplasm, cytoplasm, and nucleolus, could the authors be able to provide a physical basis of such differences based on what is known about these materials? This will be useful to support the claim that FOB microscopy can be used to study "physiological and pathological processes such as phase separation".

*Reviewer #2 (Recommendations for the authors):*

1. Maybe the authors can discuss the biological relevance of the longitudinal modulus measurement at such high frequencies

2. The authors can also discuss their choice of the diffraction tomography method. It is known that the traditional off-axis holography with lase illumination suffers from phase instability and speckles. In recent years, common-path and white light-based methods have been developed to alleviate these issues.

*Reviewer #3 (Recommendations for the authors):*

One inherent concern is that the Brillouin scattering depends heavily on the density of the medium. Therefore, while the in vitro samples can be measured in a controlled buffer, cellular milieu will add significant level of "background noise" that prevents measuring precise difference between different compartments of cells. For example, this may have produced very similar parameter for P525L-FUS vs. cytoplasm.

Another major concern, as stated above is the excessive perturbation required to perform the whole cell measurement, which the authors reported resulted in cell death.

The proof of principle was tested on what the authors call "phantom cell" which are polymers, PDMS and PAA. Although they were used due to the known mechanical property, they are not biological molecules and PDMS does not even contain a fluid phase. In addition, the RI measured by FOB did not match the known values. The differences seen here appears to be quite small as the authors claim, but they are significantly higher than other RI values authors measure in later samples. For example, Figure 2d the negligible difference between cytoplasm and nucleoplasm is stated to be significantly different.

Overall, the only significant difference measured seem to be in the cases of nucleoli, polyQ and lipid which are already known to form a more dense phase. Again, the lack of comparison to orthogonal method such as FRAP, fusion/diffusion kinetic makes it difficult to assess the validity of even the obvious cases.

The result obtained on sodium arsenite and P525L seem inconsistent with what is known from other literature. Again, the clear difference between these conditions (oxidative and mutant) may have been masked by the cellular density background.

[Editors' note: further revisions were suggested prior to acceptance, as described below.]

Thank you for submitting your article "Combined fluorescence, optical diffraction tomography and Brillouin microscopy" for consideration by *eLife*. Your article has been reviewed by 3 peer reviewers, and the evaluation has been overseen by a Reviewing Editor and Anna Akhmanova as the Senior Editor. The following individual involved in review of your submission has agreed to reveal their identity: Gabriel Popescu (Reviewer #2).

Essential revisions:

1) Reviewer 1 raises important questions regarding the measurements of longitudinal moduli and their biological relevance. Please address this explicitly.

2) The reviewer raises other important issues, specifically regarding the limitations of the FOB method that need to be addressed more thoroughly.

Please respond to this reviewer's concerns, and please do so fully and thoroughly. I agree that the title of the manuscript should be changed. As it stands, the average reader of *eLife* is not likely to know what the combination of techniques are intended for in the biological context. Please specify the intended applications in the title.

*Reviewer #1 (Recommendations for the authors):*

In the original manuscript, this reviewer and other reviewers raised important questions regarding two critical points.

The first point was: how measuring longitudinal modulus at GHz frequencies helps to understand the biological process as the authors state in their manuscript (line 439-441 in the manuscript). The authors made minimal attempt to address this, and in fact, they now have included a brief discussion on the limitations of Brillouin microscopy. There are some interesting studies in the literature which the authors cite in their argument, but that is not sufficient to make a case of how measuring longitudinal modulus at GHz frequencies helps to understand the biological process. Therefore, the response to this point remains inadequate.

The second point was pertaining to the understanding of the physics of sub-cellular mechanics. To this end, I had a question (point # 13) in the first round of review. The response again in point # 13 is superficial in my opinion and needs better considerations and/or new data.

Another important point is that the authors should include a subsection in the discussion pertaining to the limitations of FOB and data interpretation. For example, the argument presented in point # 3 should be included under the "limitations" section. The same goes for point # 7. As stated multiple times in their response letter, if we are not measuring physiologically relevant moduli and no clear conclusions can be made about the physical interpretation of the data, I share the same concern about reviewer 2 regarding how big of an impact this study will have in the field. The Nat Methods paper (Prevedel et. al 2019) that the authors cite articulates the limitations.

Overall, I am a bit surprised by the brevity of the responses by the authors from the first round of review and do not think that the revision rigorously addressed the reviewers' concerns. I also don't think the changes made in the revised manuscript are properly marked (I looked at the PDF generated by the system).

An important point which I missed in the first round of review: The title of the paper reads strange. The combination of techniques in this context is done to address a set of key biological questions, and the combination itself sounds very technical as the title of the paper.

*Reviewer #2 (Recommendations for the authors):*

The authors addressed my concerns thoroughly. I look forward to this publication in print.

*Reviewer #3 (Recommendations for the authors):*

The authors responded to all of criticisms appropriately.

---

## [Author Response]

Essential revisions:1) One of the reviewers asked for a deeper discussion of the biological relevance of the longitudinal modulus measurement at high frequencies. Are these high frequencies likely to be of importance? Perhaps in analogy with THz spectroscopy that is useful for probing the frequency dependence of dielectric responses, these high frequency longitudinal moduli might be of import for querying the elastic and / or dielectric responses. Please add a relevant discussion.

We added a section discussing the relevance of Brillouin microscopy for biological questions to the introduction. Please also see our answers to issue #8 of reviewer number 1 and to the public review of reviewer number 2 for details.

2) One of the reviewers asks for a deeper, technical discussion of the choice of the diffraction tomography method, specifically the off-axis holography with laser illumination approach. Are there issues of phase instability and speckles and how are these alleviated? Addition of data in the SI to address this point would be very helpful.

Please see our comment to the respective point raised by reviewer 2. In short, ODT and Brillouin share the same illumination source, in order to measure the refractive index (RI) and Brillouin shift at the same wavelength and not be influenced by possible dispersion. Since Brillouin requires a light source with a very narrow linewidth and, hence, a high temporal coherence length, speckle noise for ODT results. However, ODT still achieves a RI uncertainty of 4.15 × 10^-5^ (corresponding to a difference in protein concentration of 0.22 mg/ml, Kim, K. and Guck, J. Biophys. J. 119, 1946– 1957 (2020)), which is more than sufficient to pick up the RI differences in the study at hand.

We added a section regarding this issue to the discussion.

3) Please add a discussion to address the concern of independent calibration methods raised by reviewer 3. The issue is two-fold: How well does the method work with samples / condensates that are of lower density? The reconstituted G3BP1/2 based stress granule system, already studied in the Alberti lab, would be a good target because the protein is only 7-8 fold more concentrated in the stress granule facsimiles. Additionally, the addition of FRAP data as a comparative standard would be useful.

We addressed the concerns of reviewer 3 and now better reference and discuss publications showing independent reference measurements. Refractive index and protein concentration measurements in G3BP1 using ODT have been done and published by our group in Guillén-Boixet, J. et al. RNA-Induced Conformational Switching and Clustering of G3BP Drive Stress Granule Assembly by Condensation. Cell 181, 346-361.e17 (2020). We found a refractive index of ~1.35 and a protein concentration of 65 mg/ml for G3BP1 droplets. We furthermore confirmed protein concentration measurements done with ODT by volume-based measurements in McCall, P. M. et al. Quantitative phase microscopy enables precise and efficient determination of biomolecular condensate composition. bioRxiv 2020.10.25.352823 (2020) doi:10.1101/2020.10.25.352823. We do not see any particular reason why samples of lower density would cause problems, since e.g. the cytoplasm of HeLa cells exhibits densities comparable to the G3BP1 system and the PAA of the cell phantom measured shows even lower values.

FRAP measurements in polyQ aggregates have been shown by us in Kroschwald, S. et al. Promiscuous interactions and protein disaggregases determine the material state of stress-inducible RNP granules. *eLife* 4, e06807 (2015). We found that polyQ aggregates show a solid-like behavior and did not recover from photo-bleaching.

4) In addition, please make revisions to address the series of points raised by reviewer 1. And please make these changes in a way that is readily visible.

We addressed all points raised by reviewer 1 (see our comments to the respective points).

Reviewer #1 (Recommendations for the authors):In this interesting study by Schlüßler et al., the authors combined Brillouin microscopy with ODT and epi-fluorescence imaging and applied the same to study physical properties of biological materials including nucleoplasm, cytoplasm, phase-separated organelles, and adipocytes. The results are largely convincing and offer interesting insights into the material properties of these materials. There is a critical need for new methodologies to study the physical properties of biomolecular condensates in living cells under normal and pathological conditions. Hence, the current study can be significant. However, I have noted several points after carefully reading the manuscript, which is described below.1. Abstract: "Investigating the cell nucleus, we find that it has lower density but higher longitudinal modulus" compared to what exactly?

Thank you very much for this comment. The quoted sentence is indeed worded inaccurately. We changed it to now read:

“Investigating the nucleoplasm of wild-type HeLa cells, we find that it has lower density but higher longitudinal modulus than the cytoplasm.”

2. Abstract: The authors state that they have measured "absolute density with molecular specificity". Is this true? The discussion and conclusion section (p14, lines # 417-427) nicely summarizes the potential limitations of the current approach and offers future strategies on the same issue to improve upon the current foundation.

While we think that “molecular specificity” is not wrong, we agree that it might be misleading (after all, we did not measure the density of single molecules). We adjusted the respective sentence in the abstract to now read “absolute density with specificity to fluorescently labeled structures” and adjusted the usage of “molecular specificity” elsewhere.

3. P2. Line #55 – 63. The authors build their case of employing a new integrative approach (FOB microscopy) for correct longitudinal modulus measurements, however, it is not quite clear how simplifications such as using measured mass density with a priori knowledge of RI, or obtaining RI values from separate measurements lead to erroneous results of longitudinal modulus.

Using an a priori value for the RI might lead to a systematic error of the longitudinal modulus calculated, in case the assumed RI deviates from the RI actually present in the sample. Furthermore, if the RI distribution is not homogenous, neglecting the RI contribution will result in a relative error between the compartments analyzed. Whether or not this will lead to wrong trends and conclusions depends on how large the differences of the Brillouin shift and RI between the different compartments actually are (which remains unknown if not measured). For the wildtype HeLa cells measured in the manuscript, assuming a constant value for the RI would still give the correct trend for the longitudinal modulus, i.e. the modulus of the nucleoplasm would still be higher than for the cytoplasm. However, in case the differences in Brillouin shift would not be as pronounced, neglecting possible differences of the RI might conceal differences in the longitudinal modulus actually present (or introduce differences in other cases). Moreover, in case RI and density are not coupled by the two-substance mixture model (e.g. for lipid droplets in adipocytes), RI and density might not cancel out trivially, which can then also lead to systematic errors of the longitudinal modulus if the RI is neglected.

In the introduction, we now give a short summary of the reasons for deviating values of the longitudinal modulus when using these simplifications:

“These simplifications may result in an inaccurate calculation of the longitudinal modulus, since the RI distribution might not be homogenous throughout the sample, RI and density might not be coupled, hence, not cancel out, or the sample preparations necessary for separate RI measurements could influence the RI measured.”

4. In my opinion, the description of the results obtained in this study by the application of FOB microscopy as stated at the end of the introduction section is rather qualitative and can be improved.

We now give quantitative values for the RI and longitudinal modulus of cytoplasm, nucleoplasm and nucleoli in HeLa cells and a quantitative value for the difference of RI and modulus between poly-Q aggregates and cytoplasm. However, for the full list of values measured, we think the Results section is better suited.

5. Images of RI in Figure 2b, 3b, 4b and d have concentric rings throughout the images. What are these rings? If the RI is calculated from the image, then are these rings artifacts of image processing? Please explain.

The artefacts are parasitic interference patterns originated from the protective window in front of the imaging sensor of the ODT camera. This is inevitable due to the coherent nature of the laser illumination. The artifacts have a low amplitude (∆*n* < 0.002) and are averaged out during the quantitative analysis calculating the mean RI of each component.

6. Page 7. The authors state that the Brillouin shift of the nucleoplasm is "significantly" different than the cytoplasm. I have a question regarding the use of the word "significantly" here. The authors show a moduli difference of 0.038 GPa. While this may be significant in the absolute sense, it's may not be a huge difference relative to the measured moduli of the two materials. The same thing could be said for the refractive index differences, but in that case, the range of known refractive indices is small. So, please explain.

Throughout the manuscript, “significant(ly)” is now meant in the statistical sense. To make this clear, we now use “statistically significant(ly)” and changed the other four occurrences of “significant” to “substantially”.

7. The authors show that the RI and the density of the cytoplasm and nucleoplasm are comparable to water. I am curious to see what value of the longitudinal modulus will they find if they measure the same for pure water samples. For the correct modulus to be obtained, wouldn't the Brillouin shift be very different from that of the cytoplasm and nucleoplasm? In such a case, is it reasonable to assume water values for the density and the refractive index and measure only the Brillouin shift to obtain the moduli?

The longitudinal modulus of a pure water sample is around 2.2 GPa with a Brillouin shift of approximately 7.4 GHz, which is substantially different to the values measured for cytoplasm (2.410 GPa and 7.811 GHz) and nucleoplasm (2.448 GPa and 7.872 GHz) in HeLa cells. For HeLa cells, it would indeed be sufficient to assume water values for density and RI to yield the correct trend for the longitudinal modulus. However, in this case, a systematic underestimation of the longitudinal modulus would result and, obviously, the differences in density and RI of the different compartments would remain unknown.

We think that this point is already discussed sufficiently in the discussion (page 15, line 446 ff.): “We find that in all samples measured here where the two-substance mixture model can be applied, neglecting the contribution of RI and density to the longitudinal modulus still yields a similar tendency for the longitudinal modulus and Brillouin shift (i.e. a high Brillouin shift means a high longitudinal modulus and vice versa), but doing so might lead to a systematic error for the longitudinal modulus.”

8. The authors state that "These findings imply that membrane-less compartments formed by phase separation […] in spite of the thermodynamic instability inherent in this state" This information is nice and needed. But the authors need to explain why one should care about this longitudinal modulus, what does it represent, and how does it affect the material properties? In other words, the meaning of this modulus and its relation to the mechanical behavior of these materials is not clear. A discussion is enough to address this point.

The longitudinal modulus is a measure for the compressibility of the sample. It is measured by evaluating the frequency shift of incident photons due to the interaction with longitudinal photons (acoustic excitations) intrinsic to the sample. Since cells consist of liquids (water) mainly, and liquids are generally very hard to compress, the longitudinal modulus of biological samples is very high (in the GPa range) when compared to the elastic modulus. Multiple publications empirically found a correlation of the longitudinal modulus to the elastic modulus (Young’s modulus). However, this correlation does not necessarily imply a causal relation between trends of the two moduli.

We enhanced the introduction and now briefly introduce the physical meaning of the longitudinal modulus and mention the correlation to the elastic modulus. We added the following section:

“The longitudinal modulus characterizes the compressibility of a sample and is in the GPa range for common biological samples (Prevedel et al., 2019). […] However, multiple studies found empirical correlations between the longitudinal modulus and the Young’s modulus (Scarcelli et al. (2011, 2015); Schlüßler et al. (2018)).”

Since previous publications already discussed this in more detail, we do not think that an extensive review would be constructive here (see e.g. Prevedel et al. “Brillouin microscopy: an emerging tool for Mechanobiology”. Nature Methods. 2019; 16(10):969–977. doi: 10.1038/s41592-0190543-3).

9. On the previous point: The authors need to comment on these extremely large longitudinal moduli, as reported for polyQ aggregates and others. Previous studies showed that the elastic modulus of the ECM is about 1 kPa (additionally see table 1 in this report: https://www.ncbi.nlm.nih.gov/pmc/articles/PMC4553184/), so what does a value of 1 GPa for the longitudinal modulus mean? a good discussion of the meaning of the longitudinal modulus and its relation to the material properties and other moduli will be helpful.

Please see our answer to the previous point #8.

10. Page 10, lines 265-268; I see that "significantly" is used here to describe statistical significance, and maybe it's used in the same sense in the other cases too. However, the authors need to specify as it can be easily misunderstood as a significant difference in magnitude and not in statistical similarity.

We now use “statistically significant(ly)” wherever appropriate (see our answer to question number 6).

11. Page 10, lines 270-272; I am having trouble in understanding why the authors are using the standard error of the mean as opposed to the standard deviation as a measure of uncertainty. This gives an impression that the technique has superior accuracy. the variability shown in the plots is much larger than these errors (orders of magnitudes). Is this variability coming from the cell-to-cell and condensate to condensate variation or is it coming from the measurement itself? This data is not shown for the phantom cell controls (please include it in the SI). If this variability is present in that system as well, the authors may try a different system for the control to assess the inherent error of the technique, such as an aqueous two-phase system (e.g. peg and dextran). These systems are also closer to the intracellular fluid, so they may serve as better control. This would help the readers assess the accuracy of the technique itself. Alternatively, the authors should give a convincing argument for why are the SEMs a better measure of the variability of the measurement (or the uncertainty)? It's also crucial to understand the source of the variability in the refractive indices and moduli. If it is coming from the instrument, then SEMs should not be used. If it is coming from the cell-to-cell variation, then the authors may need to show data on a control system that does not have variations so that the readers can understand the level of accuracy of these measurements. If it is indeed from cell-to-cell variation and not from the experimental technique, then the use of SEMs is reasonable.

The uncertainties for the measurements in cells (HeLa wild-type, FUS stress granules, polyQ and adipocytes) are given as SEM because the single values averaged are the result from measurements in different cells. Therefore, the use of SEMs is reasonable, as the uncertainty is due to cell-to-cell variation.

However, the error given for the cell phantom measurement should actually be given as the standard deviation instead of the SEM, since only data from a single cell phantom is shown in the manuscript and the uncertainties are due to the setup and not due to variations of the sample measured. This was intended to demonstrate the accuracy of the technique itself.

We added the following sentence to the respective section “The material properties of the two components of the phantom are expected to be homogeneous so that the standard deviation (SD) of the values measured can be used as an estimate of the setups measurement uncertainty.” and corrected the uncertainty values given for the phantom measurement, so that the accuracy of the setup can now be judged correctly. Thank you very much for pointing out this problem.

12. Are the errors in the case of FUS and polyQ condensates estimated from different condensates in the same cell or from one condensate in multiple cells? this is important to judge the accuracy of this measurement and the possible effect of reducing the ROI, would reducing the ROI of the Brillouin experiment on polyQ aggregates give the same result?

The errors are given as SEM and result from measurements of a single compartment/stress granule or aggregate per cell. In total, we measured 22 different cells in case of polyQ aggregates and 90 different cells in case of FUS stress granules. We added a sentence to clarify this in the FUS stress granule section:

“In total, we measured 90 different cells, with the number of cells per compartment and condition varying between 13 and 22, as shown in Figure 4e and f.”

Reducing the region-of-interest (ROI) of the Brillouin measurements of polyQ aggregates might slightly change the quantitative values measured, in case only a specific region of the aggregate is targeted (e.g. the edge), since the finite spatial resolution of the setup leads to lower Brillouin shifts close to the surface of the aggregates. However, we would consider this a biased measurement and we prevent this by discarding measurement values close to the interface between aggregate and periphery when segmenting the polyQ aggregates. Hence, the results shown should not be affected by the chosen ROI.

13. Regarding the difference in the RI and longitudinal moduli of the nucleoplasm, cytoplasm, and nucleolus, could the authors be able to provide a physical basis of such differences based on what is known about these materials? This will be useful to support the claim that FOB microscopy can be used to study "physiological and pathological processes such as phase separation".

Unfortunately, we do not currently have a model that would explain the differences in RI and longitudinal moduli of the different compartments in HeLa cells. As written in the manuscript, we do think that the chromatin network of the nuclei has an influence on the measured values (see Liu L, Plawinski L, Durrieu MC, Audoin B. Label-Free Multi-Parametric Imaging of Single Cells: Dual Picosecond Optoacoustic Microscopy. Journal of Biophotonics. 2019; 12(8):e201900045. doi: 10.1002/jbio.201900045.), but a verified conclusion would require substantially more work on this question.

However, in order to further support the claim that FOB can be used to study "physiological and pathological processes such as phase separation", we now highlight the possibility of Brillouin microscopy to probe not only longitudinal phonons but also transverse phonons in the sample. Transverse phonons are related to the shear modulus of the sample and cannot propagate in liquids. Thus, measuring transverse phonons could help to discriminate liquid-like versus solid-like materials and analyze phase separations and transitions.

We added this section to the discussion:

“Furthermore, the setup could be enhanced to measure not only longitudinal phonons, but also transverse phonons, which are related to the shear modulus and can generally not propagate in liquids. Hence, this could help to discriminate liquid-like versus solid-like materials.”

Reviewer #2 (Recommendations for the authors):1. Maybe the authors can discuss the biological relevance of the longitudinal modulus measurement at such high frequencies.

Please see our comment to issue #8 raised by reviewer 1. The mentioned relations have been described already in previous publications such as the excellent review by Prevedel et al. “Brillouin microscopy: an emerging tool for Mechanobiology”. Nature Methods. 2019; 16(10):969–977. doi: 10.1038/s41592-019-0543-3 and an extensive discussion is not exactly the scope of the manuscript at hand. However, in order to briefly discuss the biological relevance and to give further references to the interested reader, we enhanced the introduction by the following section:

“The longitudinal modulus characterizes the compressibility of a sample and is in the GPa range for common biological samples (Prevedel et al., 2019). […] However, multiple studies found empirical correlations between the longitudinal modulus and the Young’s modulus (Scarcelli et al. (2011, 2015); Schlüßler et al. (2018)).”

2. The authors can also discuss their choice of the diffraction tomography method. It is known that the traditional off-axis holography with lase illumination suffers from phase instability and speckles. In recent years, common-path and white light-based methods have been developed to alleviate these issues.

As the reviewer pointed out, the traditional off-axis holography with laser illumination encounters phase instability and speckles. Nonetheless, the refractive index precision of the current ODT system is achieved as 4.15 × 10^-5^ (corresponding to a difference in protein concentration of 0.22 mg/ml), which is measured from the standard error of the time series of RI tomograms (Kim, K. and Guck, J. Biophys. J. 119, 1946–1957 (2020)). Hence, this measurement uncertainty is sufficient to pick up the RI differences between the various regions of the cell characterized in the study at hand.

Although common-path interferometry or shearing interferometry can enhance the phase stability, ODT using common-path interferometry requires additional descanning of scattered fields with complicated optical setups (Kim, Y. et al. Sci. Rep. 4, 6659 (2014), Chowdhury, S. et al., Optica 4, 537 (2017)), and ODT using shearing interferometry has limited field-of-view and requires low sample density in order to fulfill the imaging condition (Kim, K. et al. Opt. Lett. 39, 6935 (2014)., Guo, R. et al., Biomed. Opt. Express 12, 1869 (2021)). Hence, we stick to the MachZehnder interferometry with a proper enclosure to measure a large field-of-view using a simple optical setup with still reasonable phase stability and RI precision.

Using temporally incoherent illumination including white light or supercontinuum laser can reduce the speckle noise in digital holographic microscopy. However, in this study, ODT should use the same laser beam for Brillouin microscopy to correlate the RI value and Brillouin frequency shift of the samples at the same wavelength. For that reason, ODT uses the same laser beam for Brillouin microscopy, which requires extremely narrow laser linewidth, therefore, extended temporal coherency. The speckle noise in the current ODT configuration can be reduced by other methods such as using a dynamic diffuser (Choi, Y. et al., Opt. Lett. 36, 2465 (2011)) or numerical filtering approaches (Bianco, V. et al. Light Sci. Appl. 7, 48 (2018)).

In the revised manuscript, we added this section to the discussion:

“To correlate the RI value and Brillouin frequency shift of the samples at the same wavelength, the FOB setup uses the same laser source for ODT and Brillouin mode. […] The speckle noise could be further reduced by using a dynamic diffuser (Choi, Y. et al., Opt. Lett. 36, 2465 (2011)) or numerical filtering approaches (Bianco, V. et al. Light Sci. Appl. 7, 48 (2018)).”

Reviewer #3 (Recommendations for the authors):One inherent concern is that the Brillouin scattering depends heavily on the density of the medium. Therefore, while the in vitro samples can be measured in a controlled buffer, cellular milieu will add significant level of "background noise" that prevents measuring precise difference between different compartments of cells. For example, this may have produced very similar parameter for P525L-FUS vs. cytoplasm.

Indeed, measuring technical reference samples such as methanol or in vitro samples in controlled environments generates substantially less noisy data than measuring biological samples in vivo*,* primarily due to cell-to-cell variability. However, this limitation affects virtually every measurement technique and can be overcome by averaging multiple samples and measurements.

Furthermore, the confocality of the Brillouin microscopy setup ensures that photons interacting with the culture medium do not reach the Brillouin spectrometer and only photons, which are Brillouin scattered in the probed sample volume contribute to the measured Brillouin shift. In addition, the combination of Brillouin microscopy and ODT explicitly enables to measure the density of the probed sample volume so that differences of the local cell density are taken into account when calculating the longitudinal modulus. Hence, variations in the local cell density do not lead to “background noise” that would conceal differences of the longitudinal modulus between different cell compartment. This can be seen by the shown differences of the longitudinal modulus and refractive index between different compartments (cytoplasm, nucleoplasm and nucleoli) and polyQ aggregates in HeLa cells which could be extracted by the presented setup.

Another major concern, as stated above is the excessive perturbation required to perform the whole cell measurement, which the authors reported resulted in cell death.

Thank you for this comment. It is true that due to the low scattering efficiency of Brillouin scattering, Brillouin microscopy needs high illumination powers (e.g. when compared to Confocal fluorescence microscopy or ODT). Typically, we use 10 mW of laser power at the laser focus. As reported in the manuscript, HeLa cells with GFP-labeled FUS reacted with cell death when a large area of the cell, especially when including the nucleus, was exposed to laser illumination. Therefore, we adjusted the laser scanning scheme for these cells in a way that only a small area of the cell (the GFP-FUS stress granules and an adjacent area of the cytoplasm) was exposed. This effectively prevented cell death. Other cell types (wild-type HeLa, polyQ transfected wild-type HeLa and adipocytes) were not affected by the laser illumination. Hence, the data shown in the manuscript does not include values from cells suffering from cell death.

We adjusted the respective section to make this clearer. The section now reads:

“We therefore did not acquire Brillouin shift maps of complete P525L HeLa cells, but only of small regions of 5 µm by 5 µm around the SGs or the corresponding regions in the cytoplasm of the control cells. […] Hence, all measurements presented here stem form living cells.”

Furthermore, photo toxicity is highly dependent on the energy of the photons the sample is exposed to, hence, the wavelength of the laser used. Less energetic photons cause substantially less photo toxicity. To demonstrate this, we added a comparison of P525L HeLa GFP-FUS cells before and after full-cell and local Brillouin measurements with the FOB setup at 532 nm and a stand-alone Brillouin microscope using a wavelength of 780 nm, respectively. As shown in the Figure 4—figure supplement 1 in the revised manuscript, a scan of the full cell with a wavelength of 532 nm causes cell death. In difference, a measurement localized to a small area within the cytoplasm or stress granule, as it is performed in the manuscript for P525L HeLa cells, does not cause cell death. Furthermore, when using an excitation wavelength of 780 nm, also a full-cell scan does not cause any visible damage or cell death. Hence, cells presented in the manuscript are not damaged by the measurements and cell death is not a principle effect of the proposed method, but only of the wavelength of the laser used here. This laser was selected, because it is known to work well for ODT and requires no further technical effort for cleaning the laser emission spectrum for Brillouin microscopy illumination. Future FOB setups will use a higher wavelength of 780 nm as it is successfully used for Brillouin microscopy already by our stand-alone Brillouin setup (Schlüßler et al. “Mechanical Mapping of Spinal Cord Growth and Repair in Living Zebrafish Larvae by Brillouin Imaging”. Biophysical Journal. 2018) and is known to work for ODT (see Shin, S. et al. Common-path diffraction optical tomography with a low-coherence illumination for reducing speckle noise. Proc. SPIE 9336, 933629, 2015).

The proof of principle was tested on what the authors call "phantom cell" which are polymers, PDMS and PAA. Although they were used due to the known mechanical property, they are not biological molecules and PDMS does not even contain a fluid phase. In addition, the RI measured by FOB did not match the known values. The differences seen here appears to be quite small as the authors claim, but they are significantly higher than other RI values authors measure in later samples. For example, Figure 2d the negligible difference between cytoplasm and nucleoplasm is stated to be significantly different.

While it is true that we find a systematic difference (underestimation) of the refractive index of PDMS measured by FOB compared to the literature value measured with a different technique, this systematic difference does not invalidate or influence comparisons between data sets solely acquired by FOB. This is because all FOB measurements would be affected by the same systematic difference, which then cancels out when comparing FOB data to FOB data. Hence, the reported statistically significant differences are valid even though FOB might have a systematic difference to literature data larger than the measured differences between cell compartments.

Furthermore, as discussed in the manuscript, the systematic difference of the refractive index of PDMS measured with FOB might be explained by a swelling of the PDMS beads during the fabrication process, which could reduce the refractive index.

Overall, the only significant difference measured seem to be in the cases of nucleoli, polyQ and lipid which are already known to form a more dense phase. Again, the lack of comparison to orthogonal method such as FRAP, fusion/diffusion kinetic makes it difficult to assess the validity of even the obvious cases.

Thank you for this comment. Indeed, the verification of values measured with Brillouin microscopy is an important yet difficult issue to address, since no other technique we know of measures the longitudinal modulus of cells in vivo. However, Brillouin microscopy has been shown to be sensitive to e.g. actin polymerization, and correlations of the longitudinal modulus to the elastic modulus have been found. Hence, orthogonal techniques such as AFM have been used to show the validity of the results acquired by Brillouin microscopy. Furthermore, protein concentration measurements by ODT have been verified by a volume-based measurement approach by our group (McCall et al. “Quantitative Phase Microscopy Enables Precise and Efficient Determination of Biomolecular Condensate Composition” bioRxiv, 2020 Oct; p. 2020.10.25.352823. doi: 10.1101/2020.10.25.352823). FRAP measurements have also been done already on polyQ aggregates by our group and published in *eLife* (see Kroschwald et al. “Promiscuous Interactions and Protein Disaggregases Determine the Material State of Stress-Inducible RNP Granules” *eLife*. 2015 Aug; 4:e06807. doi: 10.7554/*eLife*.06807.). These measurements show that polyQ aggregates show solid-like properties, as the fluorescence intensity did not recover.

We think that separately repeating these comparison measurements would not yield substantial new insight and referencing the published results should be sufficient. To make these relations more clear, we added the following sentences to the introduction:

“Protein concentrations acquired with ODT were shown to agree well with results from volume-based measurements and did not suffer from differences in the quantum yield of fluorescent dyes between dilute and condensed phase as it might happen for fluorescence intensity ratio measurements (McCall et al., 2020).”

The result obtained on sodium arsenite and P525L seem inconsistent with what is known from other literature. Again, the clear difference between these conditions (oxidative and mutant) may have been masked by the cellular density background.

As stated above, “cellular density background” is not expected to mask differences in the mechanical properties of cells measured by FOB. While the differences between our measurements in P525L HeLa cells and the literature data really seemed inconsistent at first glance (although thought to be explainable by the differences in culture conditions and cell type), we now additionally performed measurements on fixed, sodium arsenite treated cells, which resolved this discrepancy. We find that in sodium arsenite treated, fixed cells SGs show a statistically significantly higher longitudinal modulus than the surrounding peri-SG cytoplasm. This is in agreement to the literature results. We adjusted the respective section. Please refer to the manuscript section “GFP-FUS stress granules in living P525L HeLa cells show RI and longitudinal modulus similar to the surrounding cytoplasm” on page 9 ff. and especially page 12 lines 333 to 349 for the changes made.

[Editors' note: further revisions were suggested prior to acceptance, as described below.]

Essential revisions:1) Reviewer 1 raises important questions regarding the measurements of longitudinal moduli and their biological relevance. Please address this explicitly.2) The reviewer raises other important issues, specifically regarding the limitations of the FOB method that need to be addressed more thoroughly.

Please see our detailed responses to the reviewers concerns below.

Please respond to this reviewer's concerns, and please do so fully and thoroughly. I agree that the title of the manuscript should be changed. As it stands, the average reader of eLife is not likely to know what the combination of techniques are intended for in the biological context. Please specify the intended applications in the title.

We understand the concern of the reviewer and editor that the current title is too technical for the audience of *eLife*. We adjusted the title, which now reads:

“Correlative all-optical quantification of mass density and mechanics of sub-cellular compartments with fluorescence specificity”

Reviewer #1 (Recommendations for the authors):In the original manuscript, this reviewer and other reviewers raised important questions regarding two critical points.The first point was: how measuring longitudinal modulus at GHz frequencies helps to understand the biological process as the authors state in their manuscript (line 439-441 in the manuscript). The authors made minimal attempt to address this, and in fact, they now have included a brief discussion on the limitations of Brillouin microscopy. There are some interesting studies in the literature which the authors cite in their argument, but that is not sufficient to make a case of how measuring longitudinal modulus at GHz frequencies helps to understand the biological process. Therefore, the response to this point remains inadequate.

The physiological relevance of the longitudinal modulus in the GHz range, i.e. whether cells are sensitive or react to changes of the high-frequency compressibility, is indeed still under debate and not yet proven. We agree with Prevedel et al. that this is “a fact that needs to be considered in the biological interpretation of results obtained by Brillouin microscopy” and think that further investigations are necessary to either demonstrate or refute the relevance of Brillouin microscopy for biology. Because of this, we strongly believe that studies such as ours help to delineate what can be detected by Brillouin microscopy and contribute to the scientific enterprise of uncovering the unknown (rather than only confirming what is known). Having said this, there are already several studies implying biological relevance of the longitudinal modulus:

– Phenomenological correlations between the Young’s and the longitudinal modulus have been found e.g. for cells under osmotic shock, for porcine and bovine lenses, for hydrogels and in zebrafish tissue. This indicates that for these samples the longitudinal modulus could serve as a proxy of the Young’s modulus, which is widely accepted as relevant for biology. See Scarcelli et al. 2011, 2015; Schlüßler et al. 2018.

– The longitudinal modulus of cells changes due to inhibition of actin polymerization with cytochalasin D or latrunculin-A, and increases with increasing substrate rigidity (Scarcelli et al. 2015; Antonacci, Braakman 2016). The modulus first decreases and subsequently increases with progressing recovery after spinal cord injury in zebrafish tissue (Schlüßler et al. 2018). Hence, various biophysical influences were shown to affect the longitudinal modulus.

– UV-induced crosslinking of polymer and actin polymerization into a gel are accompanied by an increase of the longitudinal modulus measured. See Scarcelli et al. 2008; Scarcelli et al. 2015.

It is clear that these observations do not prove that cells or organisms experience (as in: are sensitive to and respond to) changes of the longitudinal modulus. However, the measured differences of the longitudinal modulus are physically real and could influence biophysical processes directly, even without the cells or organisms being able to sense these differences in the first place. For example, variations in the longitudinal modulus at GHz frequencies detected in cells could be reporting on local changes in intermolecular interactions, water mobility and hydration shells, and other aspects relevant for emergent, supramolecular processes, which are important in their own right.

Furthermore, liquid-to-solid phase transitions not only affect the longitudinal modulus, but can also lead to variations of the samples’ viscosity or shear modulus. These quantities can also be detected by Brillouin microscopy – by evaluating the peak linewidth and detecting transverse phonons (which cannot propagate in liquids), respectively. Hence, the longitudinal modulus is not the only material property provided by Brillouin microscopy that could be used to observe phase transitions in cells. See Antonacci et al. 2018; Kim et al. 2016b; Prevedel et al. 2019.

Most of these points have already been comprehensively discussed in previous dedicated publications and are summarized well e.g. in Prevedel et al.. However, in order to enable a critical assessment of the setup presented in the manuscript for readers not familiar with Brillouin microscopy, we added a section covering these points to the discussion:

“Although Brillouin microscopy was introduced to biology more than a decade ago (Scarcelli and Yun, 2008), its relevance to biological questions is sometimes still viewed skeptically in the field of mechanobiology. […] We thus believe that Brillouin microscopy can contribute to open questions in biology, but further studies are necessary to finally establish its relevance.”

We further added an evaluation of the linewidths of the different HeLa cell compartments and found a statistically significantly higher linewidth in the nucleoli compared to the cyto- and nucleoplasms.

We also slightly toned down the mentioned conclusion of the manuscript to now read:

“This enables quantitative measurements of the mechanical properties of single cells and their compartments and might potentially contribute to a better understanding of physiological and pathological processes such as phase separation and transition in cells as a response to external stress.”

The second point was pertaining to the understanding of the physics of sub-cellular mechanics. To this end, I had a question (point # 13) in the first round of review. The response again in point # 13 is superficial in my opinion and needs better considerations and/or new data.

As a reminder, the original question asked in the first round of review was:

“Regarding the difference in the RI and longitudinal moduli of the nucleoplasm, cytoplasm, and nucleolus, could the authors be able to provide a physical basis of such differences based on what is known about these materials? This will be useful to support the claim that FOB microscopy can be used to study "physiological and pathological processes such as phase separation".”

To the authors’ best knowledge there are no publications available that analytically or computationally model the longitudinal modulus of the cytoplasm or nucleus of cells in the GHz range, similar to how it was done for elastic modulus data in the kHz range acquired e.g. by atomic force microscopy, micropipette aspiration or constricted migration (see Hobson, C. M. and Stephens, A. D. Modeling of Cell Nuclear Mechanics: Classes, Components, and Applications. Cells 9, 1623 (2020) for a good overview of the modeling of cell nuclear mechanics). Hence, theoretical insight into the behavior of the longitudinal modulus is still rather limited.

However, the nucleus has been described as a “network of chromatin and other intranuclear components surrounded by a cytosol-like fluid” (Wachsmuth et al., Anomalous diffusion of fluorescent probes inside living cell nuclei investigated by spatially-resolved fluorescence correlation spectroscopy. Journal of Molecular Biology 298, 677–689, 2000) and can display rather unusual mechanical properties, for example having a negative Poisson ratio (Pagliara et al., Auxetic nuclei in embryonic stem cells exiting pluripotency. Nature Materials 13, 638–644, 2014). Later publications attributed the nucleus’ mechanical response in the GHz frequency range to the chromatin network (Zouani et al., Universality of the network-dynamics of the cell nucleus at high frequencies. Soft Matter 10, 8737–8743, 2014) and implied that the mass density in the cytoskeleton network could be higher than in the chromatin network although the Brillouin shift behaves in the opposite way (Liu et al., Label-free multi-parametric imaging of single cells: dual picosecond optoacoustic microscopy. Journal of Biophotonics 12, e201900045, 2019).

It is also not required that mass density scales with longitudinal modulus. As a Gedankenexperiment, a non-crosslinked polymer network, with crosslinking molecules already present in the mix but not engaged, will have the same mass density, but lower elastic modulus, compared to the same network with the crosslinking molecules engaged. The uncoupling of mass density from elastic modulus is even more easily imaginable for a topologically constrained polymer network such as the nuclear chromatin. The topological constraint restricts the packing density, which leads to relatively low mass density, but the polymer network can still have a relatively robust elastic resistance to deformation or compression.

Hence, we think that the chromatin network in the nucleoplasm could very well explain the higher longitudinal modulus of the nucleoplasm compared to the cytoplasm, even though the mass density in the nucleoplasm is lower than in the cytoplasm. Future measurements could further test this explanation by e.g. probing the response of the longitudinal modulus to chromatin decondensation due to trichostatin A (TSA) treatment.

We amended the discussion accordingly:

“The nucleus is described as a "network of chromatin and other intranuclear components surrounded by a cytosol fluid" by Wachsmuth et al. (2000). […] Further analysis, e.g. testing the response of the longitudinal modulus to chromatin (de)condensation, can be performed to further consolidate this idea.”

Another important point is that the authors should include a subsection in the discussion pertaining to the limitations of FOB and data interpretation. For example, the argument presented in point # 3 should be included under the "limitations" section. The same goes for point # 7. As stated multiple times in their response letter, if we are not measuring physiologically relevant moduli and no clear conclusions can be made about the physical interpretation of the data, I share the same concern about reviewer 2 regarding how big of an impact this study will have in the field. The Nat Methods paper (Prevedel et. al 2019) that the authors cite articulates the limitations.

Unfortunately, we cannot fully follow the reviewer’s intention here. The arguments presented in points #3 and #7 do not discuss limitations of the FOB setup. These points rather demonstrate that the assumptions necessary when the refractive index is not directly measured (hence, when doing standalone Brillouin microscopy instead of FOB) can lead to systematic errors of the longitudinal modulus (although still yielding similar tendencies for Brillouin shift and modulus). These systematic errors of the longitudinal modulus do not occur when using the FOB setup. Hence, we think that this is an argument to the advantage of FOB microscopy, which is already sufficiently discussed in the introduction and discussion of the manuscript.

However, the FOB setup has limitations that we cover in the discussion:

– True molecular specificity including the absolute concentration of different molecules is not feasible with fluorescence imaging only, but can be achieved by Raman scattering microscopy. This enhancement could be added to the FOB microscope in a straightforward way.

– The Brillouin microscopy modality requires relatively long acquisition times in the range of 100 ms to 1 s per point with illumination powers in the 10 mW range. In combination with the currently used green laser source, this leads to phototoxic effects for cells. Moving to near-infrared would reduce the phototoxicity and allow higher illumination powers, hence, reducing the acquisition time.

– The refractive index measurement with ODT only works for weakly scattering samples such as single cells or beads. Thicker samples such as tissues and organisms cannot be measured with the current implementation. However, future improvements of the tomogram reconstruction taking into account multiple scattering could overcome this.

– The biological relevance of the longitudinal modulus we discuss in a separate section of the discussion now.

Overall, I am a bit surprised by the brevity of the responses by the authors from the first round of review and do not think that the revision rigorously addressed the reviewers' concerns.

We certainly did not take the reviewers’ comments lightly, as they were of relevance, helpful and interesting. Nor do we think we superficially discussed the concerns raised. If brevity provides convincing clarification of the points raised, why elaborate? After all, we covered all but two questions from the first round of review to the satisfaction of this reviewer, and all concerns of the other two reviewers. The remaining two points under discussion were very profound and very difficult to answer, especially considering that they are also heavily debated within the Brillouin microscopy community as a whole. Now, with this second opportunity for improvement, we hope that we also provided satisfactory clarifications and amendments to settle them for the purpose of this present manuscript.

An important point which I missed in the first round of review: The title of the paper reads strange. The combination of techniques in this context is done to address a set of key biological questions, and the combination itself sounds very technical as the title of the paper.

We understand the concern of the reviewer and editor that the current title is too technical for the audience of *eLife*. We adjusted the title, which now reads:

“Correlative all-optical quantification of mass density and mechanics of sub-cellular compartments with fluorescence specificity”